# DecomPose: Disentangling Cross-Category Optimization Contention for Category-Level 6D Object Pose Estimation

**Yifan Gao** [1 2]  **Lu Zou** [1]  **Zhangjin Huang** [2]  **Guoping Wang** [3]

## Abstract

Category-level 6D object pose estimation is typically formulated as a multi-category joint learning problem with fully shared model parameters. However, pronounced geometric heterogeneity across categories entangles incompatible optimization signals in shared modules, resulting in gradient conflicts and negative transfer during training. To address this challenge, we first introduce gradient-based diagnostics to quantify module-level cross-category contention. Building on results of diagnostics, we propose DecomPose, a difficulty-aware decomposition framework that mitigates optimization contention via: (1) difficulty-aware gradient decoupling, which groups categories using a data-driven difficulty proxy and routes each instance to a group-specific correspondence branch to isolate incompatible updates; and (2) stability-driven asymmetric branching, which assigns higher-capacity branches to structurally simple categories as stable optimization anchors while constraining complex categories with lightweight branches to suppress noisy updates and alleviate negative transfer. Extensive experiments on REAL275, CAMERA25, and HouseCat6D demonstrate that DecomPose effectively reduces cross-category optimization contention and delivers superior pose estimation performance across multiple benchmarks.

## 1. Introduction

Object pose estimation aims to recover the 3D rotation and translation of objects from visual observations. Existing approaches are broadly categorized into instance-level and category-level settings (Liu et al., 2026). Instance-level methods (Peng et al., 2019; He et al., 2021; Castro & Kim, 2023; Nguyen et al., 2024) assume access to object-specific CAD models with known scales and estimate a rigid 6DoF pose, but rely on precise geometric priors and exhibit limited generalization to unseen objects. In contrast, category-level methods target novel instances without prior shape models. Given the inherent intra-class variation in shape and size, this task necessitates scale estimation, thereby targeting a 9DoF pose comprising rotation, translation, and scale.

Most existing category-level pose estimation methods follow a correspondence-based paradigm (Wang et al., 2019), where sparse or dense correspondences between observed points and canonical category templates are constructed and then used to recover object pose. Although substantial progress has been made through improved feature representations (Chen & Dou, 2021; Wang et al., 2021) or stronger geometric constraints (Zhang et al., 2022; Lin et al., 2022), a persistent performance imbalance across categories is still widely observed. As shown in Figure 1(a), models trained under a unified multi-category setting typically perform well on geometrically simple categories but struggle on categories exhibiting complex structures or large intra-category shape variation. This systemic imbalance suggests that the limitations of existing methods stem not only from representation capacity, but also from the dynamics of how heterogeneous categories interact during joint optimization.

Motivated by this observation, we shift the focus from modeling intra-category variation to understanding and mitigating cross-category interactions in joint learning. Conventional frameworks employ a unified architecture with a shared backbone and task heads, forcing structurally heterogeneous categories to compete within the same parameter space. From an optimization perspective, full parameter sharing makes cross-category contention inevitable. As illustrated in Figure 1(b), this contention arises from two mechanisms. First, mismatched granularity requirements lead to gradient conflicts: complex categories demand fine-grained geometric modeling, while simple categories are adequately represented with coarse features, causing their gradients to push shared parameters in conflicting directions. Second, asynchronous convergence induces negative trans-

---

[1] Hubei Key Laboratory of Intelligent Robot, Wuhan Institute of Technology, Wuhan, Hubei, China [2] University of Science and Technology of China, Hefei, Anhui, China [3] Peking University, Beijing, China . Correspondence to: Lu Zou <lzou@wit.edu.cn>.

*Proceedings of the $43^{rd}$ International Conference on Machine Learning*, Seoul, South Korea. PMLR 306, 2026. Copyright 2026 by the author(s).

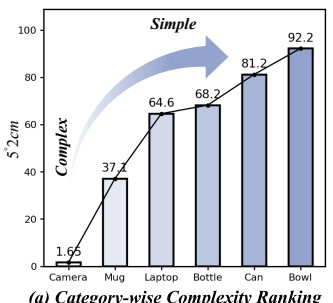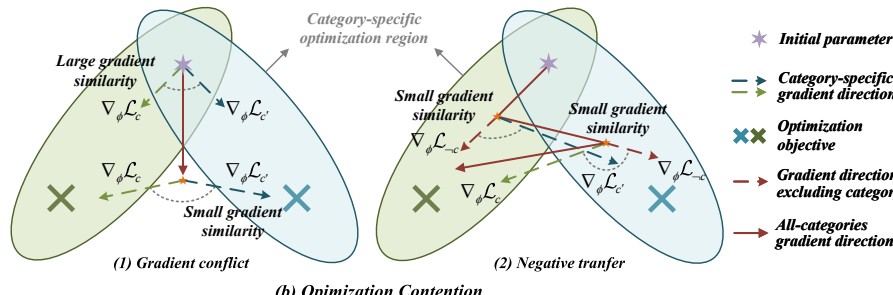

*(a) Category-wise Complexity Ranking*  
*(b) Opimization Contention*

*Figure 1.* Motivation of DecomPose. (a) Category-wise complexity ranking derived from AG-Pose (Lin et al., 2024) evaluation scores, ordering categories from complex to simple using the $5°2$cm metric. (b) Cross-category optimization contention: mismatched modeling demands lead to gradient conflicts, while asynchronous convergence induces negative transfer, as gradients from hard categories continually perturb parameters preferred by easy ones during training.

fer: simple categories converge early, whereas complex categories remain difficult to fit and continue generating large gradients, repeatedly perturbing the parameters preferred by simple categories and degrading their performance.

To better characterize this challenge, we introduce gradient-based diagnostics that make cross-category optimization contention measurable at the module level, revealing that the correspondence module is the dominant source of interference. Building on this observation, we propose Decom-Pose, a minimal yet structured intervention that selectively decouples the major source of contention and renders multi-category optimization more controllable. DecomPose routes categories to group-specific correspondence branches and assigns lightweight branches to harder groups and high-capacity branches to easier ones to improve optimization stability. As illustrated in Figure 1(c), this design preserves shared representation learning in the backbone and pose head while substantially reducing gradient conflicts and negative transfer. Our main contributions are threefold:

- We formulate cross-category optimization contention in category-level pose estimation as a measurable bottleneck, and introduce gradient-based diagnostics that quantify both category-to-category conflicts and category-to-others negative transfer.

- We propose DecomPose, a minimal yet structured intervention that mitigates cross-category contention by selectively decoupling correspondence learning through difficulty-aware routing and asymmetric branch capacity, while keeping the backbone and pose head shared.

- Extensive experiments on CAMERA25, REAL275, and HouseCat6D demonstrate that DecomPose improves category-wise pose estimation accuracy and mitigates cross-category optimization contention, achieving state-of-the-art performance.

**Conflict of Interest Disclosure** The authors declare that they have no conflicts of interest related to this work, includ-

ing no financial, commercial, or personal relationships that could reasonably be perceived as influencing the results or interpretation of this study.

## 2. Related Work

### 2.1. Dense Correspondence-Based Methods

Dense correspondence-based methods estimate object pose by establishing point-wise correspondences between camera-space observations and object-centric canonical coordinates, followed by geometric fitting. NOCS (Wang et al., 2019) introduces the Normalized Object Coordinate Space to map diverse object instances into a shared canonical frame, enabling category-level pose estimation, while subsequent works such as SPD (Tian et al., 2020) and its extensions (Chen & Dou, 2021; Lin et al., 2022; Zhang et al., 2022) incorporate deformable shape priors to better handle intra-category variation. To reduce reliance on fixed priors, more recent approaches adopt flexible correspondence formulations, including learnable query-based matching (Wang et al., 2023) and implicit transformation-based alignment (Liu et al., 2023). Despite their effectiveness, dense correspondence-based methods remain sensitive to noise, occlusion, and incomplete observations, making reliable point-wise matching challenging in practice.

### 2.2. Sparse Keypoint-Based Methods

To alleviate the limitations of dense correspondence estimation, recent studies increasingly adopt sparse keypoint-based representations. By predicting a compact set of structurally representative points, these methods reduce computational complexity and improve robustness to noise and occlusion. AG-Pose (Lin et al., 2024) introduces an instance-adaptive keypoint detection paradigm that dynamically selects representative points to better handle intra-category shape variation. CleanPose (Lin et al., 2025) further addresses data bias by modeling keypoints as mediators within a causal

framework, leveraging causal inference (Pearl, 2009; Yao et al., 2021) and knowledge distillation (Hinton et al., 2015; Gou et al., 2021) to suppress spurious correlations.

While these methods improve robustness to intra-category variation, they are typically trained under a fully shared multi-category optimization paradigm. By treating structurally heterogeneous categories as a unified objective, cross-category interactions during training are largely overlooked, which can lead to gradient interference and negative transfer. In contrast, our work adopts an optimization-centric perspective and addresses this issue by decoupling heterogeneous optimization dynamics across categories, thereby reducing cross-category interference and enabling more robust multi-category joint modeling.

## 3. Preliminaries

### 3.1. Problem Definition

Category-level object pose estimation aims to predict the full 9DoF pose of unseen object instances from a predefined category set $\mathcal{C}$. For an object instance, we construct an object-centric observation $\mathcal{O} \triangleq (\mathcal{I}, \mathcal{P}, c)$, where $\mathcal{I} \in \mathbb{R}^{H \times W \times 3}$ is the cropped RGB image, $\mathcal{P} \in \mathbb{R}^{N \times 3}$ is the point cloud in the camera frame, and $c \in \mathcal{C}$ is its category label. For a given category $c$, let $\mathcal{D}_c \subseteq \mathcal{D}$ denote the subset of samples with label $c$, and let $\mathbb{E}_{\mathbf{O} \sim \mathcal{D}_c}[\cdot]$ denote the empirical average over $\mathcal{D}_c$. We partition the parameters into blocks $\{\psi, \phi, \omega\}$, corresponding to the backbone $b_\psi$, the correspondence module $h_\phi$, and the pose head $p_\omega$. This decomposition helps isolate cross-category optimization interference and design targeted architectural interventions.

### 3.2. Cross-Category Gradient Interaction

We characterize cross-category optimization behavior by analyzing gradient interactions for each parameter block $\theta \in \{\psi, \phi, \omega\}$. For each category $c \in \mathcal{C}$, we denote the average gradient on block $\theta$ as

$$\mathbf{g}_c \triangleq \nabla_\theta \, \mathbb{E}_{\mathbf{O} \sim \mathcal{D}_c} \big[ \mathcal{L}(\mathbf{O}; \theta) \big]. \qquad (1)$$

We quantify cross-category gradient interactions by measuring two quantities: category-to-category interaction, $\mathcal{S}_{\mathrm{cc}}(c, c')$, and category-to-all interaction, $\mathcal{S}_{\mathrm{ca}}(c)$. The former quantifies the cosine similarity between the gradients of two categories, $c$ and $c'$, for a given parameter block $\theta$. The latter measures the alignment of category $c$'s gradient with the aggregated gradients of all other categories.

**Category-to-Category Interaction.** We quantify the interaction between categories $c$ and $c'$ for block $\theta$ using cosine similarity:

$$\mathcal{S}_{\mathrm{cc}}(c, c') \triangleq \cos(\mathbf{g}_c, \mathbf{g}_{c'}) = \frac{\langle \mathbf{g}_c, \mathbf{g}_{c'} \rangle}{\|\mathbf{g}_c\|_2 \|\mathbf{g}_{c'}\|_2}. \qquad (2)$$

Larger values indicate more compatible update directions, while smaller values indicate stronger disagreement.

**Category-to-All Interaction.** To summarize the interaction of category $c$ with the remaining categories, we define the aggregate gradient for block $\theta$ as

$$\mathbf{g}_{\neg c} \triangleq \mathbb{E}_{c' \in \mathcal{C} \setminus \{c\}} \big[ \mathbf{g}_{c'} \big], \qquad (3)$$

where the aggregation is performed over the gradients of all categories except for $c$. The term $\mathbb{E}$ represents the average of the gradients $\mathbf{g}_{c'}$ for all other categories $c' \in \mathcal{C} \setminus \{c\}$. Aggregation can be done with uniform or data-frequency weights, depending on how contributions from different categories are treated.

We then define the category-to-all interaction score as:

$$\mathcal{S}_{\mathrm{ca}}(c) \triangleq \cos(\mathbf{g}_c, \mathbf{g}_{\neg c}) = \frac{\langle \mathbf{g}_c, \mathbf{g}_{\neg c} \rangle}{\|\mathbf{g}_c\|_2 \|\mathbf{g}_{\neg c}\|_2}. \qquad (4)$$

Smaller $\mathcal{S}_{\mathrm{ca}}(c)$ indicates that the update direction of category $c$ is less aligned with the aggregated update direction of the remaining categories.

## 4. Diagnosing Contention Localization

### 4.1. Measuring Cross-Category Contention

We quantify cross-category optimization behavior for each parameter block $\theta \in \{\psi, \phi, \omega\}$ using gradient interaction statistics. Cross-category contention is characterized by the distribution of category-to-category cosine interactions $\mathcal{S}_{\mathrm{cc}}(c, c')$ over category pairs $c \neq c'$, for which we report the mean and variance:

$$\begin{aligned} \mu_{\mathrm{cc}} &\triangleq \mathbb{E}_{c \neq c'} \left[ \mathcal{S}_{\mathrm{cc}}(c, c') \right], \\ \sigma_{\mathrm{cc}}^2 &\triangleq \mathrm{Var}_{c' \neq c} \left[ S_{cc}(c, c') \right]. \end{aligned} \qquad (5)$$

Smaller $\mu_{\mathrm{cc}}$ indicates weaker overall alignment, while larger $\sigma_{\mathrm{cc}}^2$ reflects greater heterogeneity in cross-category updates.

Negative transfer for a target category $c$ on block $\theta$ is quantified by its misalignment with the remaining categories:

$$\mathcal{N}(c) \triangleq 1 - \mathcal{S}_{\mathrm{ca}}(c), \qquad (6)$$

where larger values indicate more severe negative transfer.

### 4.2. Why Contention Concentrates on the Correspondence Module?

Motivated by gradient-based multi-task optimization analyses (Yu et al., 2020; Wu et al., 2022; Sun et al., 2020), we explain the observed module-wise contention by decomposing category-specific gradients. For a parameter block $\theta$, the gradient of category $c$ is denoted as

$$\mathbf{g}_c = \mathbf{u} + \mathbf{v}_c, \qquad (7)$$

where $\mathbf{u}$ is the shared component common across categories, and $\mathbf{v}_c$ captures category-specific deviations. Under this decomposition, the interaction scores $\mathcal{S}_{\mathrm{cc}}(c, c')$ and $\mathcal{S}_{\mathrm{ca}}(c)$ are expressed as:

$$
\begin{aligned}
\mathcal{S}_{\mathrm{cc}}(c, c') &= \frac{\|\mathbf{u}\|_2^2 + \langle \mathbf{v}_c, \mathbf{v}_{c'} \rangle}{\sqrt{\|\mathbf{u}\|_2^2 + \|\mathbf{v}_c\|_2^2}\sqrt{\|\mathbf{u}\|_2^2 + \|\mathbf{v}_{c'}\|_2^2}}, \\
\mathcal{S}_{\mathrm{ca}}(c) &= \frac{\|\mathbf{u}\|_2^2 + \langle \mathbf{v}_c, \mathbf{v}_{\neg c} \rangle}{\sqrt{\|\mathbf{u}\|_2^2 + \|\mathbf{v}_c\|_2^2}\sqrt{\|\mathbf{u}\|_2^2 + \|\mathbf{v}_{\neg c}\|_2^2}}.
\end{aligned}
\tag{8}
$$

To quantify the relative strength of these components, we define the module-wise heterogeneity ratio as

$$
r_\theta \triangleq \frac{\mathbb{E}_c\left[\|\mathbf{v}_c\|_2^2\right]}{\|\mathbf{u}\|_2^2}.
\tag{9}
$$

This ratio governs cross-category alignment. Assuming uncorrelated deviations, $\langle \mathbf{v}_c, \mathbf{v}_{c'} \rangle \approx 0$, the interaction terms in Eq. (8) scale as $\mathcal{S} \propto (1 + r_\theta)^{-1}$. As $r_\theta$ increases, category-specific deviations dominate the shared component, leading to less aligned gradients on average. This explains the observed metric degradation: a larger $r_\theta$ results in smaller $\mu_{\mathrm{cc}}$, larger $\sigma_{\mathrm{cc}}^2$, and increased negative transfer $\mathcal{N}(c)$. We provide the full derivation in Appendix A.

In category-level object pose estimation, the backbone $\psi$ and pose head $\omega$ are primarily optimized by category-invariant objectives, inducing a strong shared gradient component across categories. In contrast, the correspondence module $\phi$ learns category-specific mappings, so its gradients are dominated by category-specific deviations, yielding the lowest interaction scores and the highest contention statistics. Therefore, we expect:

$$
r_\phi \gg \{r_\psi, r_\omega\} \implies
\begin{cases}
\mu_{\phi_{\mathrm{cc}}} < \{\mu_{\psi_{\mathrm{cc}}}, \mu_{\omega_{\mathrm{cc}}}\}, \\
\sigma_{\phi_{\mathrm{cc}}}^2 > \{\sigma_{\psi_{\mathrm{cc}}}^2, \sigma_{\omega_{\mathrm{cc}}}^2\}, \\
\mathcal{N}_\phi(c) > \{\mathcal{N}_\psi(c), \mathcal{N}_\omega(c)\}.
\end{cases}
\tag{10}
$$

This indicates that optimization contention is primarily concentrated in the correspondence module, as supported by experimental results in Sec. 6.

## 5. Method of DecomPose

As indicated in Sec. 4, optimization contention is primarily concentrated in the correspondence module $\phi$, while the backbone $\psi$ and pose head $\omega$ exhibit category-invariant behaviors. To address this issue, DecomPose adopts a minimal structured decoupling strategy: only the correspondence module $\phi$ is split into group-specific branches, while the backbone and pose head remain fully shared.

Following this principle, we first introduce a shared canonical feature extractor that provides a unified representation

for all categories (Sec. 5.1). We then construct a difficulty-aware routing function that groups categories with compatible correspondence demands (Sec. 5.2). Based on these groups, we design asymmetric correspondence branches to stabilize optimization under heterogeneity (Sec. 5.3). Finally, all branch outputs are integrated by a shared pose recovery head with a single end-to-end objective (Sec. 5.4).

### 5.1. Shared Canonical Feature Extraction

We first extract a shared canonical representation that serves as the common input to all subsequent correspondence branches. Given an object-centric observation $\mathbf{O} = \{\mathcal{I}, \mathcal{P}, c\}$, we encode per-point geometric and semantic features in a unified feature space, where $c$ denotes the category of the object instance. For the point cloud $\mathcal{P} = \{\mathbf{x}_i\}_{i=1}^N$, we use a trainable PointNet++ (Qi et al., 2017) encoder to obtain dense geometric features $\mathcal{F}_P \in \mathbb{R}^{N \times D}$. For the RGB image $\mathcal{I}$, we extract image features using a frozen DINOv2 encoder (Oquab et al., 2023) and project them to each point via camera intrinsics, yielding dense semantic features $\mathcal{F}_I \in \mathbb{R}^{N \times D}$. The two streams are concatenated per point to form the shared canonical representation $\mathcal{F} = [\mathcal{F}_P, \mathcal{F}_I] \in \mathbb{R}^{N \times 2D}$. This representation is category-agnostic and is trained jointly across all categories, ensuring that low-level geometric and semantic cues remain shared and comparable, irrespective of the object's category.

### 5.2. Difficulty-Aware Grouping as Proxy Optimization

A shared representation inherently couples correspondence learning across all categories within a unified parameter space. However, when categories exhibit substantial geometric heterogeneity, such joint optimization often induces incompatible gradient directions on the shared weights, manifesting as destructive optimization contention. To mitigate this interference, we introduce a difficulty-aware grouping strategy. While this necessitates a routing function $\gamma : \mathcal{C} \to \mathcal{G}$, learning such routing online is impractical due to the instability of non-stationary stochastic gradients. Consequently, we construct $\gamma$ offline using a pre-computed difficulty proxy and keep it fixed throughout training and inference to ensure optimization stability.

**Proxy-Based Quantile Grouping.** To instantiate the routing $\gamma$, we first compute a category-specific difficulty score $d(c) = 1 - \mathcal{T}_c$ for each category $c$, where $T_c$ is the confidence score computed by a reference estimator $\mathcal{E}$ using the object-centric observation $\mathcal{O} = (\mathcal{I}, \mathcal{P}, c)$ and ground-truth pose $\mathbb{P}$. The evaluation protocol $\mathcal{T}$ is used to evaluate the performance of the estimator by comparing the predicted pose with the ground truth. A higher confidence score $\mathcal{T}_c$ indicates lower difficulty. The category's difficulty score $d(c)$ is obtained by aggregating the instance-level confidence scores within $c$ over the subset $\mathcal{D}_c \subseteq \mathcal{D}$. We then

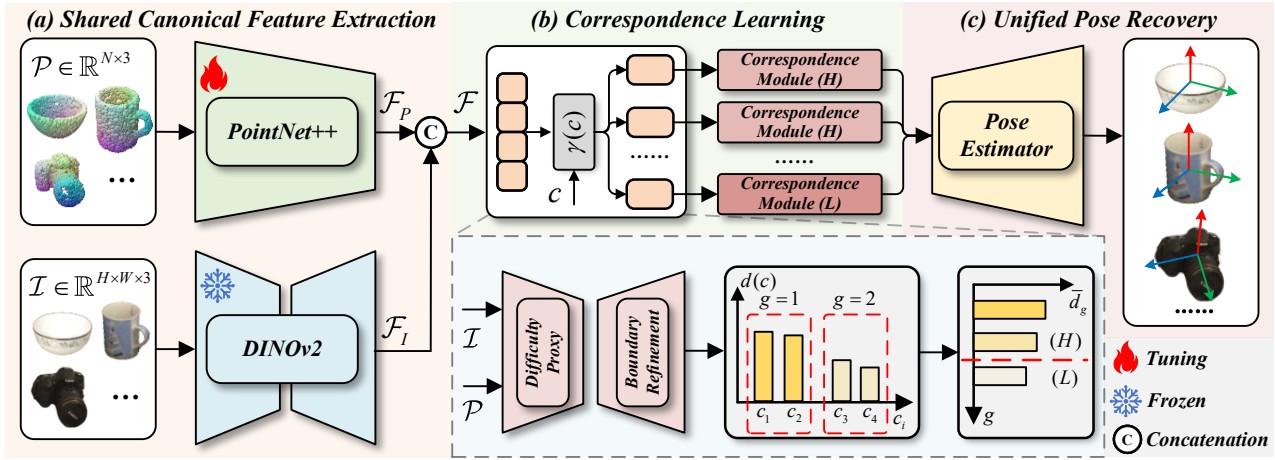

*Figure 2.* Overview of DecomPose. DecomPose adopts a minimal structured decoupling strategy in which only the correspondence learning component is partitioned into group-specific branches, while the backbone and the pose recovery head remain fully shared.

sort the categories in $\mathcal{C} = \{c_1, c_2, \ldots, c_K\}$ by their difficulty, with indices $\Pi = \{\pi(1), \pi(2), \ldots, \pi(K)\}$, such that $d(c_{\pi(1)}) \le d(c_{\pi(2)}) \le \cdots \le d(c_{\pi(K)})$.

Next, we partition $\mathcal{C}$ into $G$ difficulty groups using quantile thresholding. The boundary indices for these groups are given by $b_g = \lfloor \frac{gK}{G} \rfloor$ for $g \in \{0, \ldots, G\}$, which determines the categories assigned to each group. Specifically, the initial grouping for group $g$ is:

$$\mathcal{C}_g = \{ c_{\pi(i)} \mid b_{g-1} < i \le b_g \}. \tag{11}$$

This yields a static mapping $\gamma : \mathcal{C} \to \{1, \ldots, G\}$ where $\gamma(c) = g$ iff $c \in \mathcal{C}_g$.

**Boundary Refinement.** The rigid nature of quantile-based partitioning may arbitrarily separate categories with similar geometric complexities, particularly those situated near the group boundaries. Such arbitrary cut-offs can lead to suboptimal routing for marginal categories. To mitigate this potential misalignment, we implement a lightweight boundary sensitivity check.

For each internal boundary $g \in \{1, \ldots, G-1\}$, we identify the marginal category $c_{\pi(b_g)}$ lying on the cut-off point. We then perform a rapid pilot evaluation to compare its validation performance scores, denoted as $s_g$ and $s_{g+1}$, under the assignment of the current group $g$ and the adjacent harder group $g + 1$, respectively. Based on these pilot scores, the routing assignment $\gamma$ for the boundary category is updated to favor the configuration that yields superior stability:

$$\gamma(c_{\pi(b_g)}) \leftarrow \begin{cases} g + 1, & \text{if } s_{g+1} > s_g, \\ g, & \text{otherwise.} \end{cases} \tag{12}$$

This refinement incurs negligible computational overhead by limiting additional inference strictly to boundary categories, thereby ensuring robust routing stability.

**Algorithm 1** Difficulty-Aware Grouping

---

**Require:** Category set $\mathcal{C}$, number of groups $G$, reference estimator $\mathcal{E}$, evaluation protocol $\mathcal{T}$, object-centric observation $\mathcal{O}$, ground-truth pose $\mathbb{P}$
**Ensure:** Fixed routing $\gamma : \mathcal{C} \to \{1, \ldots, G\}$
    /* Proxy-Based Quantile Grouping */
1:  $\mathcal{T}_c \leftarrow \mathbb{E}_{\mathcal{O} \sim P(\mathcal{O}|c)}\left[\mathcal{T}(\mathcal{E}(\mathcal{O}), \mathbb{P})\right], \quad \forall c \in \mathcal{C}$
2:  $d(c) \leftarrow 1 - \mathcal{T}_c$ for $c$ in $\mathcal{C}$
3:  $\Pi \leftarrow \text{argsort}_{c \in \mathcal{C}}(d(c)) \{d(c_{\pi(1)}) \le \cdots \le d(c_{\pi(K)})\}$
4:  $b_g \leftarrow \lfloor \frac{gK}{G} \rfloor$ for $g = 0, \ldots, G$ $\{b_0 = 0, \ b_G = K\}$
5:  $\gamma(c_{\pi(i)}) \leftarrow g$ s.t. $b_{g-1} < i \le b_g, \quad \forall i \in \{1, \ldots, K\}$
    /* Boundary Refinement */
6: **for** $g = 1$ **to** $G - 1$ **do**
7:     $c^\star \leftarrow c_{\pi(b_g)}$
8:     $s_g \leftarrow \mathbb{E}_{\mathcal{O} \sim P(\mathcal{O}|c^\star)}\left[\mathcal{T}(\mathcal{E}_{\gamma(c^\star)=g}(\mathcal{O}), \mathbb{P})\right]$
9:     $s_{g+1} \leftarrow \mathbb{E}_{\mathcal{O} \sim P(\mathcal{O}|c^\star)}\left[\mathcal{T}(\mathcal{E}_{\gamma(c^\star)=g+1}(\mathcal{O}), \mathbb{P})\right]$
10:    **if** $s_g > s_{g+1}$ **then** $\gamma(c^\star) \leftarrow g + 1$
11: **end for**
12: **return** $\gamma$

---

The complete process of difficulty-aware grouping, including both proxy-based quantile grouping and boundary refinement, is outlined in Algorithm 1.

### 5.3. Stability-Driven Asymmetric Branching

**Routing and Capacity Allocation.** We decompose the correspondence learning into $G$ group-specific branches $\{h_{\phi_g}\}_{g=1}^G$. Each category $c$ routes the shared canonical features $\mathcal{F}$ exclusively to branch $h_{\phi_{\gamma(c)}}$. Crucially, we assign an asymmetric capacity $\alpha(g) \in \{\text{H}, \text{L}\}$ (High/Low) to each group based on its mean difficulty $\bar{d}_g$. Following our theoretical analysis, we employ a Stability-Driven Allocation

strategy:

$$\alpha(g) = \begin{cases} \text{H}, & \bar{d}_g < \text{Median}(\{d(c)\}), \\ \text{L}, & \text{otherwise.} \end{cases} \quad (13)$$

This design leverages high-capacity branches (e.g., complex deformation networks (Lin et al., 2024)) for easy categories to anchor the shared backbone, while constraining hard categories with lightweight branches to act as implicit regularizers against noise.

**Unified Branch Interface and Supervision.** Regardless of capacity $\alpha(g)$, all branches output a standardized set of geometric descriptors: $h_{\phi_g}(\mathcal{F}) \triangleq \{\mathcal{P}_g^{\text{kpt}}, \mathcal{F}_g^{\text{kpt}}, \hat{\mathcal{P}}_g^{\text{kpt}_{\text{nocs}}}\}$. The active branch is supervised by a composite geometric loss:

$$\mathcal{L}_g = \lambda_1 \mathcal{L}_{\text{cd}} + \lambda_2 \mathcal{L}_{\text{div}} + \lambda_3 \mathcal{L}_{\text{recon}} + \lambda_4 \mathcal{L}_{\text{nocs}}. \quad (14)$$

Specifically, $\mathcal{L}_{\text{cd}}$ is a one-sided Chamfer loss (Fan et al., 2017) minimizing the distance between predicted keypoints $\mathcal{P}_g^{\text{kpt}}$ and the object surface; $\mathcal{L}_{\text{div}}$ is a margin-based loss ensuring keypoint coverage (Lin et al., 2024); $\mathcal{L}_{\text{recon}}$ enforces consistency by reconstructing the instance point cloud from keypoints, and $\mathcal{L}_{\text{nocs}}$ applies Smooth-$\ell_1$ (Girshick, 2015) loss between predicted and ground-truth NOCS coordinates, where the ground-truth NOCS is obtained by transforming the predicted keypoints with the ground-truth pose.

### 5.4. Unified Pose Recovery and Optimization

**Pose Regression.** The outputs from the active correspondence branch are aggregated into a pose feature vector $f_{\text{pose}}$ and fed into a unified pose head $p_\omega$. The head consists of three parallel MLPs that regress the 9DoF pose parameters.

**Overall Objective.** With the fixed routing $g = \gamma(c)$, DecomPose is trained end-to-end by combining the main pose supervision $\mathcal{L}_{\text{main}}$ and the routed branch supervision $\mathcal{L}_g$ defined in Sec. 5.3. The total objective is:

$$\mathcal{L} = \lambda_{\text{main}} \mathcal{L}_{\text{main}} + \lambda_g \mathcal{L}_g, \quad (15)$$

and $\mathcal{L}_{\text{main}}$ is defined as:

$$\mathcal{L}_{\text{main}} = \|\mathbf{R} - \mathbf{R}_{\text{gt}}\|_2 + \|\mathbf{t} - \mathbf{t}_{\text{gt}}\|_2 + \|\mathbf{s} - \mathbf{s}_{\text{gt}}\|_2. \quad (16)$$

Crucially, the gradients from $\mathcal{L}_g$ back-propagate only through the active branch parameters $\phi_g$ and the shared backbone $\psi$. This effectively isolates the optimization dynamics of heterogeneous groups, preventing destructive interference while preserving shared feature learning.

## 6. Experiments

### 6.1. Experimental Setting

**Datasets.** We conduct extensive evaluations on three widely used benchmarks: REAL275 (Wang et al., 2019), CAM-

ERA25 (Wang et al., 2019), and HouseCat6D (Jung et al., 2024). REAL275 is a real-world dataset with 4.3k training RGB-D images from 7 scenes and 2.75k testing images from 6 unseen scenes, covering 6 categories with 3 instances each. CAMERA25 is a large-scale synthetic dataset containing 300k rendered RGB-D images, with 25k images from 4 held-out scenes reserved for validation, spanning 184 instances across the same 6 categories. HouseCat6D is a more complex real-world benchmark featuring 10 household categories, with 20k training frames from 34 scenes and 3k testing frames from 5 held-out scenes, covering 194 instances (about 19 per category). Characterized by dense clutter, heavy occlusion, and diverse capture conditions, it presents substantially greater challenges than the NOCS benchmarks. Its distinct category set also allows us to rigorously assess cross-domain robustness.

**Evaluation Metrics.** Following previous works (Lin et al., 2024; 2025), we evaluate the performance of our method using two metrics. For 6D pose evaluation, we report the mean Average Precision (mAP) of $n°m$cm, which measures the percentage of predictions with rotation error less than $n°$ and translation error less than $m$cm. Specifically, we adopt the thresholds 5°2cm, 5°5cm, 10°2cm, and 10°5cm. For joint 6D pose and 3D size evaluation, we report the mAP of 3D Intersection over Union (IoU$_x$), where a prediction is considered correct if the IoU between the predicted and ground-truth 3D bounding boxes exceeds a threshold $x\%$.

**Implementation Details.** For fair comparison, we use the same segmentation masks as AG-Pose (Lin et al., 2024) from Mask R-CNN (He et al., 2017) and resize cropped RGB images to $224 \times 224$. Unless noted otherwise, the feature dimension is $D$=128, each point cloud contains $N$=1024 points, and each correspondence branch predicts 96 keypoints. For difficulty-aware routing, we set the number of groups to 3 on CAMERA25 and REAL275 and 4 on HouseCat6D, and keep the routing function $\gamma$ fixed during both training and inference. The difficulty scores used to construct $\gamma$ are computed using AG-Pose as the reference estimator. For branch-specific supervision, the loss weights are $[\lambda_{\text{cd}}, \lambda_{\text{div}}, \lambda_{\text{recon}}, \lambda_{\text{nocs}}] = [3.0, 10.0, 15.0, 3.0]$, and the diversity margin is $th$=0.01. The overall objective combines the main pose loss and branch-specific loss with $\lambda_{\text{main}}$=0.6 and $\lambda_g$=1.0. We train the network using Adam (Kingma, 2014) with a triangular cyclical learning rate (Smith, 2017) from $2 \times 10^{-5}$ to $5 \times 10^{-4}$. All experiments are run on a single NVIDIA RTX 5090 GPU with a batch size of 48.

### 6.2. Experimental Results and Analysis

**Results of Contention Diagnosing.** Fig. 3(a) localizes where cross-category contention emerges by tracking gradient statistics on each parameter block $\theta \in \{\psi, \phi, \omega\}$ during

*Table 1.* Comparison with state-of-the-art methods on CAMERA25, REAL275, and HouseCat6D. Best results are highlighted in **bold** (red), and second-best results are underlined (blue). '-' indicates that results are not reported.

| Methods | Source | $IoU_{50}$ | $IoU_{75}$ | $5°2cm$ | $5°5cm$ | $10°2cm$ | $10°5cm$ |
|---|---|---|---|---|---|---|---|
| **REAL275** | | | | | | | |
| SecondPose (Chen et al., 2024) | CVPR'24 | - | - | 56.2 | 63.6 | 74.7 | 86.0 |
| AG-Pose (Lin et al., 2024) | CVPR'24 | 84.1 | 80.1 | 57.0 | 64.6 | 75.1 | 84.7 |
| GCE-Pose (Li et al., 2025) | CVPR'25 | 84.1 | 79.8 | 57.0 | 65.1 | 75.6 | 86.3 |
| MK-Pose (Yang et al., 2025) | IROS'25 | 84.0 | 80.3 | 60.8 | - | 78.0 | 84.6 |
| CleanPose (Lin et al., 2025) | ICCV'25 | - | - | **61.7** | 67.6 | 78.3 | 86.3 |
| **DecomPose** | - | **84.1** | **81.1** | 61.2 | **67.9** | **79.2** | **87.5** |
| **CAMERA25** | | | | | | | |
| AG-Pose (Lin et al., 2024) | CVPR'24 | 94.2 | 92.5 | 79.5 | 83.7 | 87.1 | 92.6 |
| MK-Pose (Yang et al., 2025) | IROS'25 | 94.1 | 92.2 | 77.9 | - | 86.1 | 91.7 |
| CleanPose (Lin et al., 2025) | ICCV'25 | 94.3 | 92.5 | 80.3 | 84.2 | 87.7 | 92.7 |
| **DecomPose** | - | **94.3** | **92.7** | **80.8** | **84.5** | **87.8** | **92.8** |
| **HouseCat6D** | | | | | | | |
| SecondPose (Chen et al., 2024) | CVPR'24 | 66.1 | - | 11.0 | 13.4 | 25.3 | 35.7 |
| AG-Pose (Lin et al., 2024) | CVPR'24 | 76.9 | 53.0 | 21.3 | 22.1 | 51.3 | 54.3 |
| GCE-Pose (Li et al., 2025) | CVPR'25 | 79.2 | **60.6** | 24.8 | 25.7 | **55.4** | 58.4 |
| CleanPose (Lin et al., 2025) | ICCV'25 | **79.8** | 53.9 | 22.4 | 24.1 | 51.6 | 56.5 |
| **DecomPose** | - | 79.4 | 56.3 | **25.4** | **25.9** | 54.2 | **58.9** |

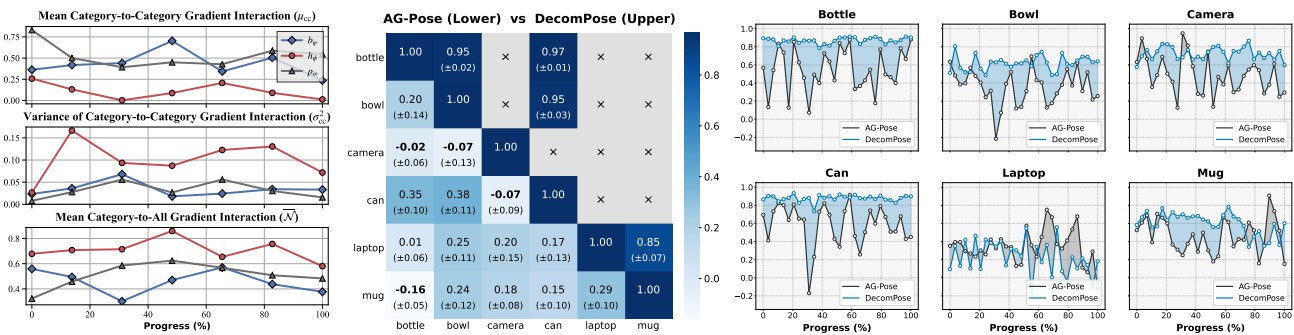

*(a)* Localization of optimization contention.
*(b)* Category-to-category gradient similarity on the module $p_\omega$.
*(c)* Category-to-all gradient similarity on the module $b_\psi$.

*Figure 3.* Visualization of cross-category gradient interactions on CAMERA25 and REAL275. (a) is computed by Eq. (2) and (b) by Eq. (4) throughout training, where smaller values indicate stronger gradient conflicts and more severe negative transfer, respectively. (c) is computed by Eq. (5), (6) throughout training, and $\overline{\mathcal{N}}$ is obtained by averaging Eq. (6) over categories. Additional visualizations on HouseCat6D are provided in Appendix F.

training. For each block, we compute the mean and variance of category-to-category cosine interactions $(\mu_{cc}, \sigma^2_{cc})$ in Eq. (5) and the category-to-all negative transfer score $\mathcal{N}(c)$ in Eq. (6), whose category-averaged form is denoted as $\overline{\mathcal{N}}$. We observe a consistent module-wise disparity: the correspondence module $\phi$ shows the smallest $\mu_{cc}$, the largest $\sigma^2_{cc}$, and the highest $\overline{\mathcal{N}}$, indicating the weakest gradient alignment, the strongest cross-category heterogeneity, and the most severe negative transfer.

**Results of DecomPose.** Table 1 summarizes the quantitative results on REAL275, CAMERA25, and HouseCat6D. Across all benchmarks, DecomPose achieves favorable results compared with the strongest recent methods and obtains the best performance on most evaluation metrics. **(1) Results on REAL275.** DecomPose achieves the best performance on the $5°5cm$, $10°2cm$, and $10°5cm$ metrics with values of 67.9, 79.2, and 87.5 respectively, surpassing the strongest CleanPose (Lin et al., 2025) by 0.3, 0.9, and 1.2. The only exception is the $5°2cm$ metric, where DecomPose reaches 61.2 and ranks second, 0.5 lower than CleanPose. **(2) Results on CAMERA25.** DecomPose achieves the best performance across all metrics. It outperforms CleanPose by 0.2 on $IoU_{75}$ and further surpasses the baseline AG-Pose (Lin et al., 2024) on the $5°2cm$, $5°5cm$, $10°2cm$, and $10°5cm$ metrics by 0.5, 0.3, 0.1, and 0.1 respectively. **(3) Results on HouseCat6D.** DecomPose outperforms GCE-

*Table 2.* Effectiveness of difficulty-aware grouping on REAL275 and HouseCat6D.

| strategy | IoU$_{50}$ | IoU$_{75}$ | $5°2cm$ | $5°5cm$ | $10°2cm$ | $10°5cm$ |
|---|---|---|---|---|---|---|
| | | | REAL275 | | | |
| None | 84.1 | 80.1 | 57.0 | 64.6 | 75.1 | 84.7 |
| Random | 84.1 | 80.3 | 59.2 | 66.4 | 76.2 | 86.4 |
| P. w/o BR. | 84.0 | 80.6 | 60.1 | 66.9 | 77.4 | 86.5 |
| P. w/ BR. | 84.1 | 81.1 | 61.1 | 67.9 | 79.3 | 87.3 |
| | | | HouseCat6D | | | |
| None | 76.9 | 53.0 | 21.3 | 22.1 | 51.3 | 54.3 |
| Random | 76.2 | 52.1 | 20.7 | 21.6 | 51.3 | 55.1 |
| P. w/o BR. | 78.6 | 54.3 | 22.9 | 23.7 | 52.6 | 57.2 |
| P. w/ BR. | 79.4 | 56.3 | 25.3 | 25.9 | 54.3 | 58.9 |

*Table 3.* Effectiveness of asymmetric correspondence branches on REAL275 and HouseCat6D.

| branch | IoU$_{50}$ | IoU$_{75}$ | $5°2cm$ | $5°5cm$ | $10°2cm$ | $10°5cm$ |
|---|---|---|---|---|---|---|
| | | | REAL275 | | | |
| L/L/L | 84.1 | 80.7 | 56.8 | 64.2 | 77.7 | 87.5 |
| H/H/H | 84.0 | 80.6 | 59.4 | 65.4 | 77.5 | 86.3 |
| L/L/H | 84.0 | 80.8 | 58.6 | 66.2 | 78.2 | 86.7 |
| H/H/L | 84.1 | 81.1 | 61.1 | 67.9 | 79.3 | 87.3 |
| | | | HouseCat6D | | | |
| L/L/L/L | 76.5 | 53.1 | 22.3 | 23.8 | 53.1 | 55.6 |
| H/H/H/H | 76.7 | 55.2 | 23.1 | 24.6 | 54.5 | 56.2 |
| L/L/H/H | 77.1 | 54.3 | 22.9 | 24.1 | 53.7 | 57.1 |
| H/H/L/L | 79.4 | 56.3 | 25.3 | 25.9 | 54.3 | 58.9 |

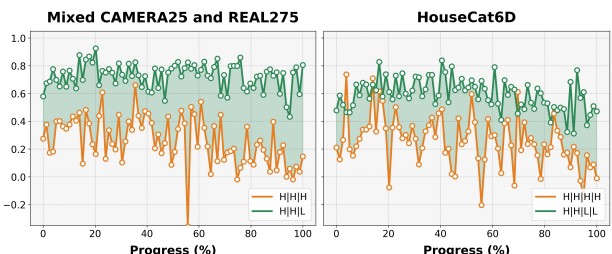

*Figure 4.* Comparison of gradient-direction dynamics on REAL275 and HouseCat6D, using the same analysis method as in Figure 3(c).

Pose (Li et al., 2025) by margins of 0.6, 0.2, and 0.5 on the $5°2cm$, $5°5cm$, and $10°5cm$ metrics respectively. Although its IoU$_{50}$ and IoU$_{75}$ rank second on HouseCat6D, the consistent improvements in pose accuracy across all three datasets are consistent with the reduction of optimization contention observed in the gradient analysis. These results further confirm the generalization capability of the proposed decomposition strategy and demonstrate its stability across diverse category sets.

We further analyze gradient interactions in Figure 3(b). The similarity matrix for the correspondence module $\phi$ shows that group-wise decoupling effectively isolates conflicting categories into separate branches, eliminating negative interactions. In contrast, AG-Pose (Lin et al., 2024) shows low or negative similarity between categories from different groups, indicating severe cross-category interference. Categories within the same branch exhibit stronger gradient alignment, indicating enhanced intra-branch optimization consistency. This is further validated in Figure 3(c), which tracks cosine alignment on the shared backbone $\psi$. Compared to the baseline, our method maintains consistently higher alignment throughout training, demonstrating reduced negative transfer and improved optimization stability.

### 6.3. Ablation Study

**Grouping by difficulty alleviates optimization contention.** Table 2 shows the effectiveness of difficulty-aware grouping. Compared with random grouping, difficulty-aware grouping

improves accuracy on REAL275 from 59.2 to 61.1 under the stricter $5°2cm$ metric and from 66.4 to 67.9 under the $5°5cm$ metric. It also improves accuracy on HouseCat6D from 20.7 to 25.3 under the stricter $5°2cm$ metric. These results indicate that the performance gains cannot be attributed solely to the introduction of grouping, but arise from effectively isolating heterogeneous categories. Figure 3(b) further supports this conclusion by visualizing reduced cross-category gradient conflicts under difficulty-aware grouping.

**Asymmetric capacity allocation benefits hard-sample learning.** To isolate the effect of branch capacity, we fix the routing function $\gamma$ and vary only the capacity of the group-specific correspondence heads. Table 3 shows that the asymmetric configurations achieve the best performance on both REAL275 and HouseCat6D, with H/H/L and H/H/L/L yielding the best results, respectively. On REAL275, assigning a lightweight branch to the hardest group improves the strict $5°2cm$ metric from 59.4 to 61.1 and IoU$_{75}$ from 80.6 to 81.1 compared with the uniform heavy-capacity configuration H/H/H. A similar trend is observed on House-Cat6D, where the asymmetric H/H/L/L setting outperforms the uniform H/H/H/H design, improving IoU$_{75}$ from 55.2 to 56.3 and the strict $5°2cm$ metric from 23.1 to 25.3. We further conduct a diagnostic analysis in Figure 4 to compare gradient-direction dynamics under uniform and asymmetric capacity allocation. The results show that the asymmetric configurations yield more stable gradient evolution, supporting our interpretation that lightweight branches act as an implicit regularizer for difficult categories. These results highlight the importance of asymmetric capacity allocation for stabilizing learning on hard samples.

**Optimization contention primarily arises in the correspondence module.** In Table 4, we keep the correspondence module $h_\phi$ shared and instead introduce three branches on either the backbone $b_\psi$ or the pose head $p_\omega$. On both REAL275 and HouseCat6D, these variants yield only marginal performance gains compared with decomposing $h_\phi$, suggesting that optimization conflicts are substantially

*Table 4.* Localization of optimization contention on REAL275 and HouseCat6D.

| Branch | IoU$_{50}$ | IoU$_{75}$ | $5°2cm$ | $5°5cm$ | $10°2cm$ | $10°5cm$ |
|---|---|---|---|---|---|---|
| | | | REAL275 | | | |
| $\{b_\psi\}$ | 84.0 | 80.9 | 57.3 | 64.9 | 75.6 | 84.6 |
| $\{p_\omega\}$ | **84.2** | 79.9 | 59.8 | 66.0 | 77.7 | 85.7 |
| $\{h_\phi\}$ | 84.1 | **81.1** | **61.1** | **67.9** | **79.3** | **87.3** |
| | | | HouseCat6D | | | |
| $\{b_\psi\}$ | 78.5 | 55.1 | 22.7 | 24.1 | 54.6 | 57.9 |
| $\{p_\omega\}$ | 78.2 | 55.8 | 23.5 | 24.3 | 54.1 | 58.2 |
| $\{h_\phi\}$ | **79.4** | **56.3** | **25.3** | **25.9** | 54.3 | **58.9** |

*Table 5.* Effectiveness of grouping numbers $G$ on REAL275 and HouseCat6D.

| $G$ | IoU$_{50}$ | IoU$_{75}$ | $5°2cm$ | $5°5cm$ | $10°2cm$ | $10°5cm$ |
|---|---|---|---|---|---|---|
| | | | REAL275 | | | |
| 1 | 84.1 | 80.1 | 57.0 | 64.6 | 75.1 | 84.7 |
| 2 | 84.1 | 80.9 | 58.5 | 65.7 | 76.9 | 86.0 |
| 3 | **84.1** | **81.1** | **61.1** | **67.9** | **79.3** | **87.3** |
| 4 | 84.1 | 80.1 | 60.5 | 66.9 | 78.1 | 86.7 |
| | | | HouseCat6D | | | |
| 1 | 76.9 | 53.0 | 21.3 | 22.1 | 51.3 | 54.3 |
| 2 | 77.6 | 54.2 | 21.9 | 24.1 | 53.4 | 55.4 |
| 3 | 77.4 | 53.9 | 22.5 | 24.3 | 52.8 | 56.8 |
| 4 | **79.4** | **56.3** | **25.3** | **25.9** | 54.3 | **58.9** |
| 5 | 78.1 | 55.6 | 23.8 | 24.5 | **55.1** | 58.1 |

*Table 6.* Robustness analysis of different reference estimators on REAL275 and HouseCat6D.

| Reference | IoU$_{50}$ | IoU$_{75}$ | $5°2cm$ | $5°5cm$ | $10°2cm$ | $10°5cm$ |
|---|---|---|---|---|---|---|
| | | | REAL275 | | | |
| IST-Net | 84.1 | 80.9 | 61.3 | 67.6 | 79.1 | **87.4** |
| SecondPose | 84.1 | 81.1 | 61.0 | **68.2** | **79.4** | 87.1 |
| AG-Pose | **84.1** | **81.1** | 61.1 | 67.9 | 79.3 | 87.3 |
| | | | HouseCat6D | | | |
| IST-Net | 78.2 | 54.9 | 23.9 | 24.7 | 53.6 | 57.8 |
| SecondPose | 79.1 | 56.5 | 24.7 | 25.4 | 53.8 | 58.6 |
| AG-Pose | **79.4** | 56.3 | **25.3** | **25.9** | **54.3** | **58.9** |

weaker in the backbone and pose head. This observation supports our design choice of decomposing the correspondence module while keeping $b_\psi$ and $p_\omega$ shared, and further indicates across both benchmarks that correspondence learning plays a primary role in cross-category optimization contention.

**Increasing the number of groups does not monotonically improve performance.** Table 5 shows that performance does not monotonically increase with the number of groups $G$. On REAL275, increasing $G$ from 1 to 3 improves most pose accuracy metrics and yields the best overall performance at $G{=}3$, whereas further increasing it to $G{=}4$ leads to performance degradation. A similar non-monotonic trend

is observed on HouseCat6D: the overall performance improves as $G$ increases from 1 to 4, but drops when $G$ is further increased to 5. This trend indicates that the gains of DecomPose mainly come from effective conflict-aware decoupling induced by an appropriate grouping structure, rather than simply increasing model capacity by introducing more branches.

**Static routing is robust to unreliable reference estimation.** Table 6 shows that our static routing strategy remains robust even when the reference estimation is not fully reliable. Specifically, we construct the difficulty proxy using different reference estimators, including IST-Net (Liu et al., 2023), SecondPose (Chen et al., 2024), and AG-Pose (Lin et al., 2024). Although different references lead to slight metric fluctuations, the overall performance remains stable across all choices. On HouseCat6D, AG-Pose provides the best overall pose accuracy, while SecondPose yields a slightly higher IoU$_{75}$. On REAL275, all three estimators deliver highly comparable performance, with only marginal differences across metrics. These results suggest that static routing does not heavily rely on a perfectly accurate reference estimator, demonstrating good robustness to imperfect difficulty estimation.

### 6.4. Conclusion

In this paper, we present DecomPose to address optimization contention in category-level 6D object pose estimation. We identify that the correspondence module is the primary bottleneck where heterogeneous category difficulties induce severe cross-category interference. To mitigate this, we propose isolating heterogeneous categories via difficulty-aware grouping and tailoring branch capacities through asymmetric allocation. Our analysis reveals that assigning lightweight branches to the hardest groups effectively improves robustness, confirming that performance gains stem from structural decoupling rather than simple model capacity expansion. Extensive experiments demonstrate that DecomPose achieves state-of-the-art performance, highlighting the effectiveness of resolving optimization conflicts for multi-category object pose estimation. One limitation of DecomPose is that its offline difficulty-based routing may become less effective as the category set grows substantially, since category difficulty provides only a coarse proxy for training hardness. More adaptive routing is an important future direction.

## Impact Statement

This paper presents work whose goal is to advance the field of Machine Learning. There are many potential societal consequences of our work, none which we feel must be specifically highlighted here.

## Acknowledgments

This work was supported by the Natural Science Foundation of Hubei Province (Grant No. 2025AFB218), the Hubei Provincial Department of Education Science and Technology Plan Project (Grant No. Q20241505), and the Scientific Research Fund of Wuhan Institute of Technology (Grant Nos. 24QD07).

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

# A. Full Derivation of Gradient Interaction Metrics

We provide full derivations for the cross-category gradient interaction metrics used in Sec. 3.1–4.2. For a parameter block $\theta \in \{\psi, \phi, \omega\}$, the category-wise average gradient is

$$\mathbf{g}_c \triangleq \nabla_\theta \, \mathbb{E}_{\mathbf{O} \sim \mathcal{D}_c} \big[ \mathcal{L}(\mathbf{O}; \theta) \big]. \tag{17}$$

The cosine interaction between two categories $c$ and $c'$ is defined as

$$S_{cc}(c, c') \triangleq \cos(\mathbf{g}_c, \mathbf{g}_{c'}) = \frac{\langle \mathbf{g}_c, \mathbf{g}_{c'} \rangle}{\|\mathbf{g}_c\|_2 \, \|\mathbf{g}_{c'}\|_2}. \tag{18}$$

For category-to-all interaction, we define the aggregated gradient of all other categories as

$$\mathbf{g}_{\neg c} \triangleq \mathbb{E}_{c' \in \mathcal{C} \setminus \{c\}}[\mathbf{g}_{c'}], \qquad S_{ca}(c) \triangleq \cos(\mathbf{g}_c, \mathbf{g}_{\neg c}) = \frac{\langle \mathbf{g}_c, \mathbf{g}_{\neg c} \rangle}{\|\mathbf{g}_c\|_2 \, \|\mathbf{g}_{\neg c}\|_2}. \tag{19}$$

## A.1. Decomposition and Exact Closed Forms

Following Sec. 4.2, we decompose the category gradient into a shared component and a category-specific deviation:

$$\mathbf{g}_c = \mathbf{u} + \mathbf{v}_c, \tag{20}$$

where $\mathbf{u}$ is a shared gradient component for the given block $\theta$, and $\mathbf{v}_c$ captures the category-dependent residual.

**Category-to-Category Interaction** $S_{cc}(c, c')$. Substituting $\mathbf{g}_c = \mathbf{u} + \mathbf{v}_c$ and $\mathbf{g}_{c'} = \mathbf{u} + \mathbf{v}_{c'}$ into the cosine similarity yields

$$\langle \mathbf{g}_c, \mathbf{g}_{c'} \rangle = \langle \mathbf{u} + \mathbf{v}_c, \mathbf{u} + \mathbf{v}_{c'} \rangle$$
$$= \|\mathbf{u}\|_2^2 + \langle \mathbf{u}, \mathbf{v}_c \rangle + \langle \mathbf{u}, \mathbf{v}_{c'} \rangle + \langle \mathbf{v}_c, \mathbf{v}_{c'} \rangle, \tag{21}$$
$$\|\mathbf{g}_c\|_2^2 = \|\mathbf{u} + \mathbf{v}_c\|_2^2 = \|\mathbf{u}\|_2^2 + 2\langle \mathbf{u}, \mathbf{v}_c \rangle + \|\mathbf{v}_c\|_2^2, \tag{22}$$
$$\|\mathbf{g}_{c'}\|_2^2 = \|\mathbf{u} + \mathbf{v}_{c'}\|_2^2 = \|\mathbf{u}\|_2^2 + 2\langle \mathbf{u}, \mathbf{v}_{c'} \rangle + \|\mathbf{v}_{c'}\|_2^2. \tag{23}$$

Therefore, the exact expression is

$$S_{cc}(c, c') = \frac{\|\mathbf{u}\|_2^2 + \langle \mathbf{u}, \mathbf{v}_c \rangle + \langle \mathbf{u}, \mathbf{v}_{c'} \rangle + \langle \mathbf{v}_c, \mathbf{v}_{c'} \rangle}{\sqrt{\|\mathbf{u}\|_2^2 + 2\langle \mathbf{u}, \mathbf{v}_c \rangle + \|\mathbf{v}_c\|_2^2} \, \sqrt{\|\mathbf{u}\|_2^2 + 2\langle \mathbf{u}, \mathbf{v}_{c'} \rangle + \|\mathbf{v}_{c'}\|_2^2}}. \tag{24}$$

We adopt a standard residual normalization that treats $\mathbf{v}_c$ as a deviation with respect to the shared component $\mathbf{u}$. Concretely, $\mathbf{u}$ can be chosen as the mean gradient over categories and $\mathbf{v}_c$ as the corresponding centered residual, which yields an approximate orthogonality

$$\langle \mathbf{u}, \mathbf{v}_c \rangle \approx 0 \quad \text{for all } c. \tag{25}$$

Under this approximation, the interaction simplifies to

$$S_{cc}(c, c') \approx \frac{\|\mathbf{u}\|_2^2 + \langle \mathbf{v}_c, \mathbf{v}_{c'} \rangle}{\sqrt{\|\mathbf{u}\|_2^2 + \|\mathbf{v}_c\|_2^2} \, \sqrt{\|\mathbf{u}\|_2^2 + \|\mathbf{v}_{c'}\|_2^2}}. \tag{26}$$

**Category-to-All Interaction** $S_{ca}(c)$. For the aggregated gradient of all other categories, we have

$$\mathbf{g}_{\neg c} = \mathbb{E}_{c' \neq c}[\mathbf{g}_{c'}] = \mathbb{E}_{c' \neq c}[\mathbf{u} + \mathbf{v}_{c'}] = \mathbf{u} + \mathbf{v}_{\neg c}, \quad \mathbf{v}_{\neg c} \triangleq \mathbb{E}_{c' \neq c}[\mathbf{v}_{c'}]. \tag{27}$$

The category-to-all interaction is then

$$S_{ca}(c) = \cos(\mathbf{g}_c, \mathbf{g}_{\neg c}) = \cos(\mathbf{u} + \mathbf{v}_c, \mathbf{u} + \mathbf{v}_{\neg c}). \tag{28}$$

Expanding as before gives

$$S_{ca}(c) = \frac{\|\mathbf{u}\|_2^2 + \langle \mathbf{u}, \mathbf{v}_c \rangle + \langle \mathbf{u}, \mathbf{v}_{\neg c} \rangle + \langle \mathbf{v}_c, \mathbf{v}_{\neg c} \rangle}{\sqrt{\|\mathbf{u}\|_2^2 + 2\langle \mathbf{u}, \mathbf{v}_c \rangle + \|\mathbf{v}_c\|_2^2} \, \sqrt{\|\mathbf{u}\|_2^2 + 2\langle \mathbf{u}, \mathbf{v}_{\neg c} \rangle + \|\mathbf{v}_{\neg c}\|_2^2}}. \tag{29}$$

Under $\langle \mathbf{u}, \mathbf{v}_c \rangle \approx 0$ and $\langle \mathbf{u}, \mathbf{v}_{\neg c} \rangle \approx 0$, we obtain

$$S_{ca}(c) = \frac{\|\mathbf{u}\|_2^2 + \langle \mathbf{v}_c, \mathbf{v}_{\neg c} \rangle}{\sqrt{\|\mathbf{u}\|_2^2 + \|\mathbf{v}_c\|_2^2} \, \sqrt{\|\mathbf{u}\|_2^2 + \|\mathbf{v}_{\neg c}\|_2^2}}. \tag{30}$$

### A.2. Dependence on the Heterogeneity Ratio $r_\theta$

We define the module-wise heterogeneity ratio

$$r_\theta = \frac{\mathbb{E}_c[\|\mathbf{v}_c\|_2^2]}{\|\mathbf{u}\|_2^2}. \tag{31}$$

For convenience, we introduce normalized deviation magnitudes

$$\alpha_c = \frac{\|\mathbf{v}_c\|_2^2}{\|\mathbf{u}\|_2^2}, \qquad \beta_c = \frac{\|\mathbf{v}_{\neg c}\|_2^2}{\|\mathbf{u}\|_2^2}, \tag{32}$$

and define the correlation coefficients between deviations as

$$\rho_{cc'} = \frac{\langle \mathbf{v}_c, \mathbf{v}_{c'} \rangle}{\|\mathbf{v}_c\|_2 \|\mathbf{v}_{c'}\|_2} \in [-1, 1], \qquad \rho_{c\neg c} = \frac{\langle \mathbf{v}_c, \mathbf{v}_{\neg c} \rangle}{\|\mathbf{v}_c\|_2 \|\mathbf{v}_{\neg c}\|_2} \in [-1, 1]. \tag{33}$$

With these definitions, the interactions can be rewritten as

$$S_{cc}(c, c') \approx \frac{1 + \rho_{cc'}\sqrt{\alpha_c \alpha_{c'}}}{\sqrt{(1 + \alpha_c)(1 + \alpha_{c'})}}, \tag{34}$$

$$S_{ca}(c) \approx \frac{1 + \rho_{c\neg c}\sqrt{\alpha_c \beta_c}}{\sqrt{(1 + \alpha_c)(1 + \beta_c)}}. \tag{35}$$

Under an uncorrelated-deviation approximation

$$\mathbb{E}[\rho_{cc'}] \approx 0, \qquad \mathbb{E}[\rho_{c\neg c}] \approx 0, \tag{36}$$

the expected interactions are dominated by the denominators. When $\alpha_c$ is concentrated around its mean and $\mathbb{E}_c[\alpha_c] \approx r_\theta$, both $S_{cc}$ and $S_{ca}$ follow a scalar scaling law of the form

$$S \propto \frac{1}{1 + r_\theta}. \tag{37}$$

Consequently, a larger $r_\theta$ implies smaller average category-to-category interaction $\mu_{cc}$, larger variance $\sigma_{cc}^2$, and larger negative transfer

$$\mathcal{N}(c) = 1 - S_{ca}(c). \tag{38}$$

This provides a module-wise indicator of how suitable a shared parameter block is for multi-category joint optimization and motivates the use of asymmetric correspondence branches in highly heterogeneous modules.

## B. Implementation Details of Gradient Diagnostics

We report implementation details for the gradient-based diagnostics used to quantify cross-category optimization contention. All diagnostics are computed on a fixed snapshot of the model during training and do not affect optimization.

### B.1. When to Collect Gradients

All gradient diagnostics are computed *offline* after the main training finishes. We save the model checkpoint at every epoch and run a one-epoch replay pass for each checkpoint. This replay pass is used solely to compute gradient statistics and does not update model parameters, i.e., no parameter updates are performed during diagnostics.

To ensure comparability across checkpoints, we construct a fixed diagnostic subset $\tilde{\mathcal{D}}$ by randomly sampling from the original training set and reuse the same subset for all checkpoints. During the replay pass, we adopt a constant learning rate that is identical for every checkpoint, eliminating the effect of learning-rate schedules on the measured gradients.

For a checkpoint at epoch $t$, we estimate category-level gradients using the fixed diagnostic subset $\tilde{\mathcal{D}}$ and a one-epoch replay pass without parameter updates.

## B.2. Category-Level Gradient Estimation

Let $\tilde{\mathcal{B}}^{(m)}$ denote the $m$-th mini-batch in the replay pass, and let $c(\tilde{\mathcal{B}}^{(m)})$ be its category label. For each parameter block $\theta_i$, we compute the mini-batch gradient

$$\mathbf{g}_{t,i}^{(m)} \triangleq \nabla_{\theta_i} \mathcal{L}\Big(\tilde{\mathcal{B}}^{(m)}; \theta^{(t)}\Big), \tag{39}$$

where $\theta^{(t)}$ denotes the parameters of checkpoint $t$. For each category $c$, we aggregate gradients over all batches belonging to $c$,

$$\hat{\mathbf{g}}_{t,c,i} \triangleq \frac{1}{|\mathcal{M}_c|} \sum_{m \in \mathcal{M}_c} \mathbf{g}_{t,i}^{(m)}, \qquad \mathcal{M}_c \triangleq \{m \mid c(\tilde{\mathcal{B}}^{(m)}) = c\}. \tag{40}$$

In addition, for each target category $c$ we estimate the gradient of "all other categories" by averaging gradients over the complement set,

$$\hat{\mathbf{g}}_{t,\neg c,i} \triangleq \frac{1}{|\mathcal{M}_{\neg c}|} \sum_{m \in \mathcal{M}_{\neg c}} \mathbf{g}_{t,i}^{(m)}, \qquad \mathcal{M}_{\neg c} \triangleq \{m \mid c(\tilde{\mathcal{B}}^{(m)}) \neq c\}. \tag{41}$$

This estimator provides a consistent reference for category-to-others interaction and negative-transfer measurements.

We keep the batch size, data order, and all diagnostic settings identical across checkpoints so that $\hat{\mathbf{g}}_{t,c,i}$ and $\hat{\mathbf{g}}_{t,\neg c,i}$ are directly comparable across $t$.

## B.3. Parameter Blocks and Gradient Vectors

We partition trainable parameters into blocks $\theta = \{\psi, \phi, \omega\}$ corresponding to the backbone, correspondence module, and pose recovery head. For each block $\theta_i$, we flatten the gradients of all tensors in the block and concatenate them into a single vector $\hat{\mathbf{g}}_{c,i} \in \mathbb{R}^{d_i}$. We exclude parameters that are not optimized and omit buffers such as BatchNorm running statistics. All cosine interactions are computed in these block-wise gradient spaces.

# C. Architecture Details of Asymmetric Correspondence Branches

## C.1. High-Capacity Correspondence Branch

The high-capacity correspondence branch adopts a keypoint-centric design to explicitly model rich local and global geometric context before regressing NOCS coordinates.

**Keypoint Extraction.**  We maintain $K$ learnable query embeddings $\mathcal{Q} \in \mathbb{R}^{K \times D}$. Given point-wise features $\mathcal{F} \in \mathbb{R}^{N \times D}$, instance-adaptive keypoint tokens are obtained via cross-attention:

$$\mathcal{Q}^{\text{ins}} = \text{Norm}\Big(\mathcal{Q} + \text{CA}(\mathcal{Q}, \mathcal{F}, \mathcal{F})\Big). \tag{42}$$

We first compute the keypoint-to-point affinity matrix:

$$\mathcal{H} = \mathcal{Q}^{\text{ins}} \mathcal{F}^\top \in \mathbb{R}^{K \times N}. \tag{43}$$

A row-wise softmax is then applied to produce the assignment weights, based on which keypoint coordinates and keypoint features are pooled:

$$\mathbf{W} = \text{softmax}(\mathcal{H}), \qquad \mathcal{P}^{\text{kpt}} = \mathbf{W}\mathcal{P} \in \mathbb{R}^{K \times 3}, \qquad \mathcal{F}^{\text{kpt}} = \mathbf{W}\mathcal{F} \in \mathbb{R}^{K \times D}. \tag{44}$$

**Local Geometry Aggregation.**  For each keypoint $i$, we retrieve $K_n$ nearest neighbors from $(\mathcal{P}, \mathcal{F})$, denoted by $\{(\mathcal{P}_{(i,j)}, \mathcal{F}_{(i,j)})\}_{j=1}^{K_n}$. Local relative offsets are encoded by

$$\Delta \mathcal{P}_{(i,j)} = \mathcal{P}_{(i,j)} - \mathcal{P}_i^{\text{kpt}}, \qquad \alpha_{i,j} = \text{MLP}\big(\Delta \mathcal{P}_{(i,j)}\big), \tag{45}$$

and summarized as a keypoint-wise descriptor

$$\mathbf{f}_i^\ell = \text{AvgPool}_j(\alpha_{i,j}). \tag{46}$$

We compute attention weights between the enriched keypoint token and its neighbors:

$$a_{i,j} = \text{sim}\Big(\text{MLP}\big([\mathcal{F}_i^{\text{kpt}}, \mathbf{f}_i^{\ell}]\big), \mathcal{F}_{(i,j)}\Big),\tag{47}$$

and aggregate local features with a residual update:

$$\mathcal{F}_i^{\text{local}} = \text{MLP}\Big(\sum_{j=1}^{K_n} \text{softmax}_j(a_{i,j})\,\mathcal{F}_{(i,j)} + \mathcal{F}_i^{\text{kpt}}\Big).\tag{48}$$

**Global Context Fusion.** We further encode keypoint-to-keypoint geometry

$$\beta_{i,j} = \text{MLP}\big(\mathcal{P}_j^{\text{kpt}} - \mathcal{P}_i^{\text{kpt}}\big), \qquad \mathbf{f}_i^{\text{global}} = \text{AvgPool}_j(\beta_{i,j}),\tag{49}$$

and compute a global pooled descriptor

$$\mathcal{F}^{\text{global}} = \text{AvgPool}_i\big(\{\mathcal{F}_i^{\text{local}}\}_{i=1}^K\big).\tag{50}$$

The final keypoint representation is given by

$$\widehat{\mathcal{F}}_i^{\text{kpt}} = \text{MLP}\big([\mathcal{F}_i^{\text{local}}, \mathcal{F}^{\text{global}}, \mathbf{f}_i^{\text{global}}]\big).\tag{51}$$

**NOCS Regression.** NOCS coordinates are predicted from $\widehat{\mathcal{F}}^{\text{kpt}}$:

$$\widehat{\mathcal{P}}^{\text{nocs}} = \text{MLP}\big(\widehat{\mathcal{F}}^{\text{kpt}}\big) \in \mathbb{R}^{K \times 3}.\tag{52}$$

### C.2. Low-Capacity Correspondence Branch

The low-capacity correspondence branch follows the same keypoint extraction procedure as above, but replaces the heavy context modeling with a lightweight geometry-biased attention mechanism.

**Keypoint Extraction.** Given $(\mathcal{P}, \mathcal{F})$, keypoint coordinates $\mathcal{P}_{\text{kpt}}$ and features $\mathcal{F}_{\text{kpt}}$ are obtained using Eqs. (42)–(44).

**Geometry-Biased Attention Aggregation.** For each keypoint $i$ and its neighbor $j$, we compute relative offsets $\Delta\mathcal{P}_{(i,j)} = \mathcal{P}_{(i,j)} - \mathcal{P}_i^{\text{kpt}}$ and map them to a compact geometric descriptor $\mathbf{g}_{i,j} = \text{MLP}\big(\Delta\mathcal{P}_{(i,j)}\big)$. A lightweight MLP predicts per-head local geometric biases:

$$\mathbf{b}_{i,j}^{\text{local}} = \text{MLP}(\mathbf{g}_{i,j}) \in \mathbb{R}^H,\tag{53}$$

where $H$ denotes the number of attention heads. We also compute a keypoint-wise global geometric bias:

$$\bar{\mathbf{g}}_i = \text{AvgPool}_j(\mathbf{g}_{i,j}), \qquad \mathbf{b}_i^{\text{global}} = \text{MLP}(\bar{\mathbf{g}}_i) \in \mathbb{R}^H.\tag{54}$$

Geometry-biased attention logits are computed as

$$\mathbf{A}_{i,j} = \text{Attn}\big(\mathcal{F}_i^{\text{kpt}}, \mathcal{F}_{(i,j)}\big) + \mathbf{b}_{i,j}^{\text{local}}.\tag{55}$$

Neighbor features are aggregated by

$$\mathcal{F}_i^{\text{local}} = \sum_{j=1}^{K_n} \text{softmax}_j(\mathbf{A}_{i,j}) \odot \mathcal{F}_{(i,j)} + \mathcal{F}_i^{\text{kpt}}.\tag{56}$$

The global geometric bias is then injected into the local feature:

$$\widehat{\mathcal{F}}_i^{\text{kpt}} = \mathcal{F}_i^{\text{local}} + \mathbf{b}_i^{\text{global}}.\tag{57}$$

**NOCS Regression.** NOCS coordinates are predicted from $\widehat{\mathcal{F}}^{\text{kpt}}$:

$$\widehat{\mathcal{P}}^{\text{nocs}} = \text{MLP}\big(\widehat{\mathcal{F}}^{\text{kpt}}\big) \in \mathbb{R}^{K \times 3}.\tag{58}$$

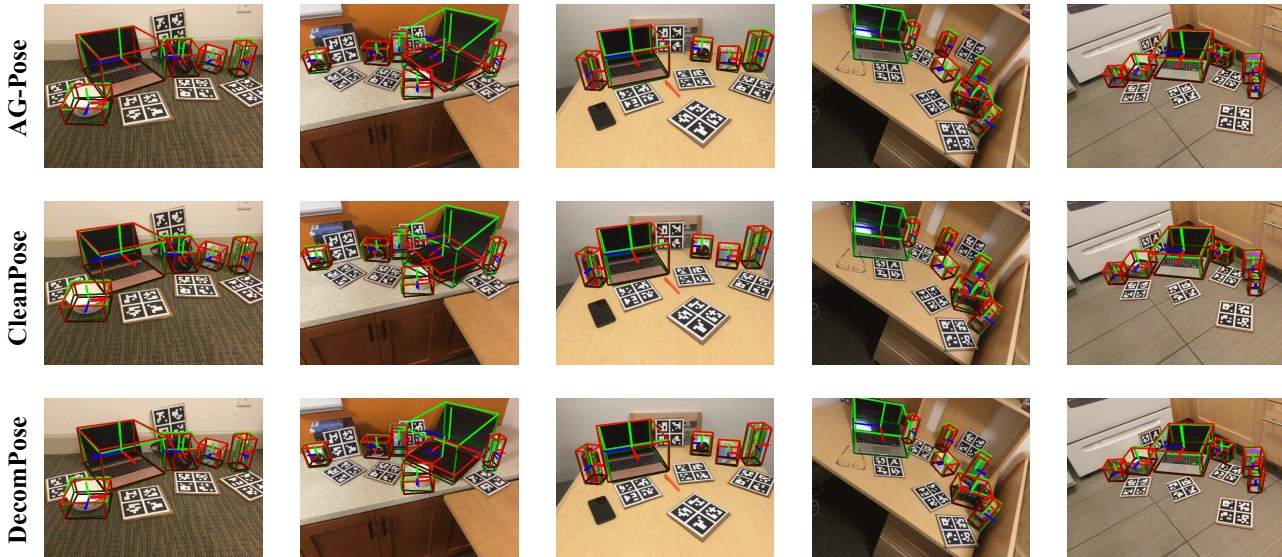

*Figure 5.* Qualitative comparisons of AG-Pose (Lin et al., 2024), CleanPose (Lin et al., 2025), and our DecomPose on REAL275. Ground truth is shown in green, and predicted results are shown in red.

## D. Qualitative Analysis

Figure 5 presents qualitative comparisons of AG-Pose (Lin et al., 2024), CleanPose (Lin et al., 2025), and our DecomPose on REAL275 across diverse real-world conditions. As shown in the first row, AG-Pose produces reasonable pose hypotheses in moderately cluttered scenes, yet exhibits clear misalignment when objects are heavily occluded, truncated, or lack discriminative texture, resulting in unstable orientations and translation drift. CleanPose (second row) improves robustness by delivering tighter alignment and reducing major failures, particularly for thin structures and symmetry-ambiguous objects. Nonetheless, noticeable errors persist under extreme occlusion or complex background interference.

In contrast, DecomPose (third row) yields more accurate and stable pose predictions across these scenarios, preserving better alignment under severe occlusion, clutter, and specular or textureless surfaces. These results demonstrate that decomposing correspondence learning alleviates cross-category interference and enhances robustness in real-world category-level 6D pose estimation.

## E. Additional Ablations

### E.1. Stability Study

To evaluate result stability, we report performance over three independent runs with different random seeds. For each dataset, the first run uses seed 1, following the AG-Pose (Lin et al., 2024) setting, and two additional runs are performed with different seeds. As shown in Table 7, the variation across runs remains consistently small for both DecomPose and AG-Pose over all datasets and evaluation metrics. In particular, on the stricter pose metrics where DecomPose achieves the most evident gains, the variation of both methods is much smaller than the performance gap between them. While three runs are still insufficient for a rigorous statistical significance test, these repeated-run results nevertheless indicate that the improvements of DecomPose over AG-Pose are stable rather than caused by incidental seed variation.

### E.2. Structural Similarity

To investigate whether the gain of difficulty-aware grouping mainly comes from grouping structurally similar categories, we further conduct an ablation study on a reduced heterogeneous subset of HouseCat6D. Specifically, we retain six relatively heterogeneous categories, including *box, remote, shoe, cutlery, teapot*, and *tube*, while excluding categories with more apparent geometric similarity. We compare three settings, i.e., *no grouping*, *random grouping*, and *difficulty-aware grouping*, under the same training protocol. As shown in Table 8, difficulty-aware grouping consistently achieves the best performance across all metrics, indicating that its advantage is not solely due to structural similarity, but also stems from grouping

*Table 7.* Stability analysis of DecomPose and AG-Pose under different random seeds on REAL275, CAMERA25, and HouseCat6D.

| Seed | Method | $IoU_{50}$ | $IoU_{75}$ | $5°2cm$ | $5°5cm$ | $10°2cm$ | $10°5cm$ |
|---|---|---|---|---|---|---|---|
| | | | REAL275 | | | | |
| 1 | AG-Pose | 84.1 | 80.1 | 57.0 | 64.6 | 75.1 | 84.7 |
| 1 | DecomPose | 84.1 | 81.1 | 61.1 | 67.9 | 79.3 | 87.3 |
| 50 | AG-Pose | 84.0 | 79.8 | 57.3 | 64.2 | 74.9 | 84.5 |
| 50 | DecomPose | 84.1 | 81.2 | 61.5 | 67.7 | 79.2 | 87.6 |
| 100 | AG-Pose | 84.0 | 80.2 | 56.8 | 64.7 | 75.4 | 84.2 |
| 100 | DecomPose | 84.1 | 81.1 | 61.0 | 68.2 | 79.0 | 87.5 |
| mean $\pm$ SEM | AG-Pose | $84.0 \pm 0.0$ | $80.0 \pm 0.1$ | $57.0 \pm 0.1$ | $64.5 \pm 0.2$ | $75.1 \pm 0.1$ | $84.5 \pm 0.1$ |
| mean $\pm$ SEM | DecomPose | $84.1 \pm 0.0$ | $81.1 \pm 0.0$ | $61.2 \pm 0.2$ | $67.9 \pm 0.1$ | $79.2 \pm 0.1$ | $87.5 \pm 0.1$ |
| | | | CAMERA25 | | | | |
| 1 | AG-Pose | 94.2 | 92.5 | 79.5 | 83.7 | 87.1 | 92.6 |
| 1 | DecomPose | 94.3 | 92.7 | 80.7 | 84.5 | 87.8 | 92.8 |
| 50 | AG-Pose | 94.2 | 92.4 | 79.3 | 83.8 | 87.2 | 92.3 |
| 50 | DecomPose | 94.3 | 92.7 | 80.8 | 84.3 | 87.9 | 92.8 |
| 100 | AG-Pose | 94.3 | 92.5 | 79.5 | 83.5 | 87.4 | 92.4 |
| 100 | DecomPose | 94.3 | 92.6 | 80.8 | 84.6 | 87.6 | 92.7 |
| mean $\pm$ SEM | AG-Pose | $94.2 \pm 0.0$ | $92.5 \pm 0.0$ | $79.4 \pm 0.1$ | $83.7 \pm 0.1$ | $87.2 \pm 0.1$ | $92.4 \pm 0.1$ |
| mean $\pm$ SEM | DecomPose | $94.3 \pm 0.0$ | $92.7 \pm 0.0$ | $80.8 \pm 0.0$ | $84.5 \pm 0.1$ | $87.8 \pm 0.1$ | $92.8 \pm 0.0$ |
| | | | HouseCat6D | | | | |
| 1 | AG-Pose | 76.9 | 53.0 | 21.3 | 22.1 | 51.3 | 54.3 |
| 1 | DecomPose | 79.4 | 56.3 | 25.3 | 25.9 | 54.3 | 58.9 |
| 50 | AG-Pose | 76.5 | 53.3 | 21.5 | 22.3 | 51.2 | 54.2 |
| 50 | DecomPose | 79.5 | 56.2 | 25.1 | 25.6 | 54.0 | 58.7 |
| 100 | AG-Pose | 76.8 | 53.2 | 20.9 | 22.4 | 51.2 | 54.6 |
| 100 | DecomPose | 79.3 | 56.5 | 25.7 | 26.1 | 54.4 | 59.2 |
| mean $\pm$ SEM | AG-Pose | $76.7 \pm 0.1$ | $53.2 \pm 0.1$ | $21.2 \pm 0.2$ | $22.3 \pm 0.1$ | $51.2 \pm 0.0$ | $54.4 \pm 0.1$ |
| mean $\pm$ SEM | DecomPose | $79.4 \pm 0.1$ | $56.3 \pm 0.1$ | $25.4 \pm 0.2$ | $25.9 \pm 0.1$ | $54.2 \pm 0.1$ | $58.9 \pm 0.1$ |

*Table 8.* Ablation study on the effect of structural similarity on the reduced heterogeneous subset of HouseCat6D.

| Method | $IoU_{50}$ | $IoU_{75}$ | $5°2cm$ | $5°5cm$ | $10°2cm$ | $10°5cm$ |
|---|---|---|---|---|---|---|
| None | 64.0 | 49.7 | 10.3 | 12.1 | 32.7 | 36.3 |
| Random | 64.4 | 50.8 | 11.2 | 13.4 | 33.5 | 36.9 |
| P. w/ BR. | **64.6** | **51.3** | **11.9** | **14.6** | **34.8** | **37.6** |

categories according to their optimization difficulty.

# F. More Diagnostic Results

In addition to the main diagnostic results, we provide further analyses on the larger HouseCat6D dataset to illustrate the effectiveness of DecomPose in mitigating cross-category optimization contention. As shown in Figure 6, across all ten categories, DecomPose exhibits more stable gradient dynamics compared to AG-Pose, indicating reduced cross-category interference and improved optimization stability.

# G. Future Work

A promising direction for future work is to develop dynamic, online routing strategies that adaptively assign categories to correspondence branches based on evolving gradient interactions, improving upon the coarse difficulty-based proxy used in DecomPose. Another exciting avenue is to integrate large pre-trained models and continual learning paradigms (Jiang et al.; 2025b;a), enabling the framework to leverage rich shared representations while incrementally incorporating new categories without inducing catastrophic cross-category interference. Extending these approaches to larger and more diverse category sets, as well as more complex real-world scenarios with heavy occlusion or multi-object interactions, would further enhance the scalability and robustness of the decomposition strategy.

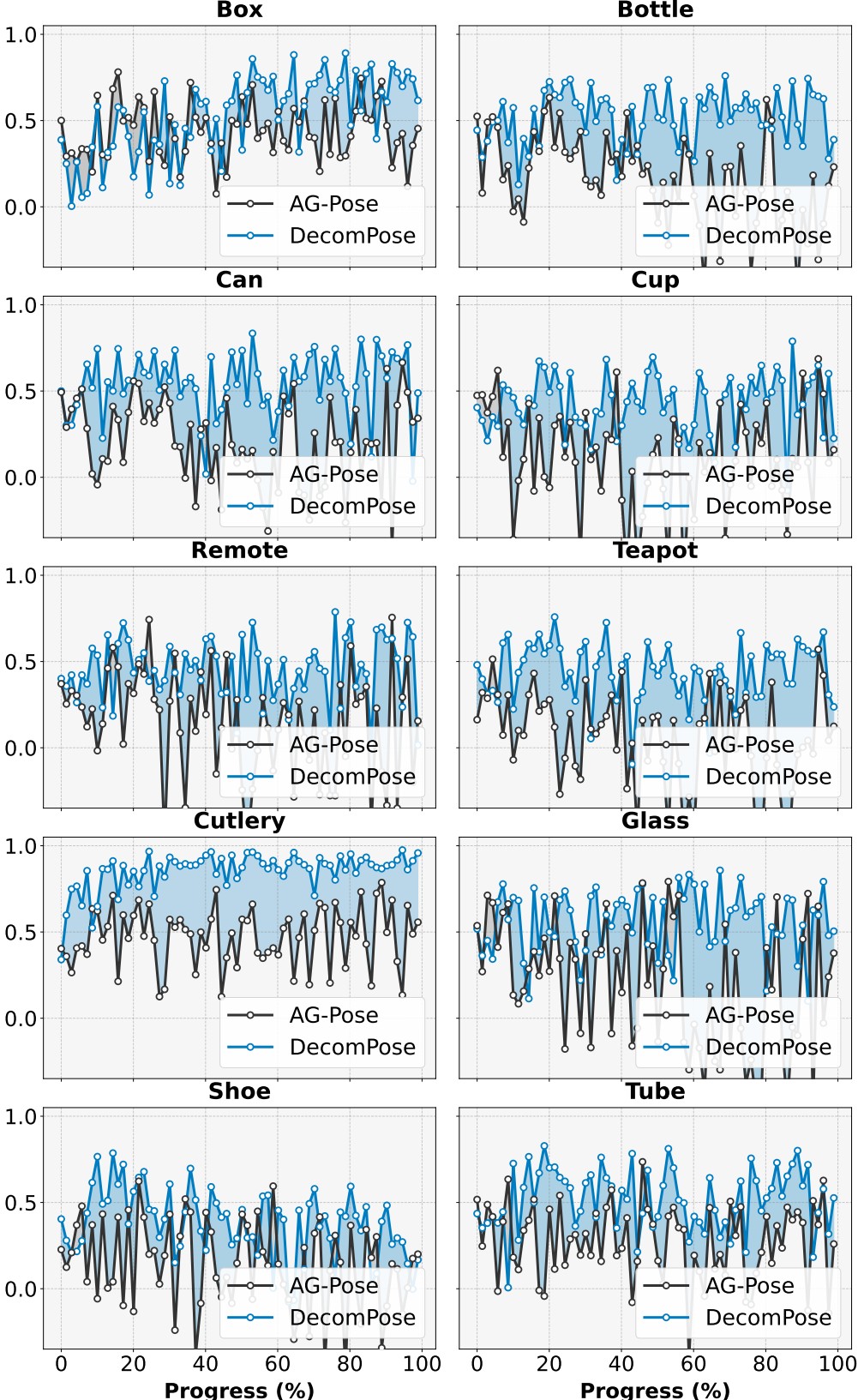

*Figure 6.* Visualization of cross-category gradient interactions on HouseCat6D.

