# OpenReview forum: "DecomPose: Disentangling Cross-Category Optimization Contention for Category-Level 6D Object Pose Estimation"
_ICML.cc/2026/Conference — ICML 2026 regular_

### Official Review · Reviewer_xFUw · 2026-02-20

**Soundness:** 2
**Presentation:** 3
**Significance:** 4
**Originality:** 3
**Overall Recommendation:** 4
**Confidence:** 4

**Summary:**

This paper reframes category-level pose estimation as an optimization problem, showing that gradient conflicts across heterogeneous categories, especially within the correspondence module, drive negative transfer in shared architectures. The authors introduce gradient-based diagnostics to reveal this contention and propose a difficulty-aware strategy with asymmetric correspondence branches, where simpler categories anchor training through high-capacity branches while harder ones are constrained to stabilize updates. Experiments on multiple datasets demonstrate consistent improvements, suggesting the gains stem from resolving cross-category interference rather than merely increasing model capacity.

**Compliance With Llm Reviewing Policy:**

Affirmed.

**Final Justification:**

The rebuttal fully addresses my questions. The justification for the math behind the framework sounds correct and provides valuable insights to leverage gradient information. I maintain my recommendation of acceptance for this paper.

**Key Questions For Authors:**

1. Why was GCE-Pose left out of the HouseCat6D comparison in Table 1? Since it was included for the REAL275 dataset in the smae section, the omission is quite noticeable. If GCE-Pose actually hits higher marks on strict metrics like $5\degree 2cm$ in complex environments, it’s hard to reconcile that with the paper's broad "state-of-the-art" claims.

2. The routing mechanism relies heavily on a single reference model (e.g. AG-Pose in this work). Different architectures should have different failure modes: some might be weak against occlusion while others struggle with symmetry. This raises a question: how much does the grouping change if the reference model is swapped? Have any studies been done using distinct baselines (for example, SecondPose or CleanPose) to see if the resulting branch assignments remain stable? Otherwise, if such stability is not a required property for the proposed method, a detailed discussion on the reasons behind should be provided.

3. The math behind the framework seems to assume that category-specific deviations are uncorrelated with the shared gradient. Early in training, gradients are notoriously noisy and unstable. To what extent does this early-stage noise impact the reliability of the gradient diagnostics shown in the training progression charts in Fig.3?

**Limitations:**

yes

**Strengths And Weaknesses:**

**Strengths**

* The paper moves beyond spatial representations to address the actual dynamics of training. By diagnosing gradient contention at the module level, it identifies the correspondence head as the main culprit for performance dips.


* The asymmetric branching is a genuinely interesting regularization trick. Bottlenecking complex categories to protect the backbone from noisy gradients while letting simple categories act as anchors is a counter-intuitive but effective way to handle geometric heterogeneity.


**Weaknesses**

* The empirical reporting for HouseCat6D feels incomplete. It omits GCE-Pose (with an $IoU_{75}$ of 60.6 as in their paper, a baseline that the authors have already presented for REAL275), which is a major competitor in complex settings, making it difficult to fully buy into the state-of-the-art claims for the specific benchmark.

* The routing system’s reliance on a frozen, offline reference estimator is a potential vulnerability. It may limits the model’s performance to the specific biases and blind spots of whatever baseline (like AG-Pose) was used for the initial difficulty proxy.

---

> ### Author Rebuttal · Authors · 2026-03-29
>
> **Response to Reviewer xFUw**
>
> We appreciate the reviewer’s helpful comments and address the concerns below.
>
> ---
>
> **W1: Missing GCE-Pose comparison on HouseCat6D in Table 1.**
>
> - Thank you for your careful reading of our paper. We have added GCE-Pose to the HouseCat6D comparison, and the updated results are shown below. After including GCE-Pose, DecomPose still achieves the best performance on three of the four pose accuracy metrics, namely $5^\circ2$cm, $5^\circ5$cm, and $10^\circ5$cm. On the remaining metrics, it also remains highly competitive, achieving second-best performance on IoU50 and IoU75. We will incorporate this updated comparison into the revised version.
>
> | HouseCat6D |  |  |  |  |  |  |
> |---|---:|---:|---:|---:|---:|---:|
> | Method | IoU50 | IoU75 | 5°2cm | 5°5cm | 10°2cm | 10°5cm |
> | SecondPose | 66.1 | - | 11.0 | 13.4 | 25.3 | 35.7 |
> | AG-Pose | 76.9 | 53.0 | 21.3 | 22.1 | 51.3 | 54.3 |
> | CleanPose | **79.8** | 53.9 | 22.4 | 24.1 | 51.6 | 56.5 |
> | GCE-Pose | 79.2 | **60.6** | 24.8 | 25.7 | **55.4** | 58.4 |
> | Ours (G=3) | 77.4 | 53.9 | 22.5 | 24.3 | 52.8 | 56.8 |
> | Ours (G=4) | 79.4 | 56.3 | **25.3** | **25.9** | 54.3 | **58.9** |
>
> ---
>
> **W2: The routing mechanism relies heavily on a single reference model.**
>
> - We replaced AG-Pose with IST-Net and SecondPose as alternative reference estimators, and the results are summarized below. We used these two baselines to test whether the routing depends critically on a specific reference model. On REAL275, where the number of categories is smaller, the induced difficulty ranking remains relatively stable across different references. On HouseCat6D, the variation is slightly larger, which is consistent with its larger category set and stronger cross-category heterogeneity, making difficulty estimation itself more challenging. In this case, earlier baselines such as IST-Net may provide coarser difficulty signals and thus lead to slightly weaker results. Overall, the performance remains broadly consistent across different reference models, suggesting that our method does not depend on a specific reference model, but rather on a reasonably consistent coarse difficulty structure.
>
> | Dataset | Reference | IoU50 | IoU75 | 5°2cm | 5°5cm | 10°2cm | 10°5cm |
> |---|---|---:|---:|---:|---:|---:|---:|
> | HouseCat6D | IST-Net | 78.2 | 54.9 | 23.9 | 24.7 | 53.6 | 57.8 |
> | HouseCat6D | SecondPose | 79.1 | **56.5** | 24.7 | 25.4 | 53.8 | 58.6 |
> | HouseCat6D | AG-Pose | **79.4** | 56.3 | **25.3** | **25.9** | **54.3** | **58.9** |
> | REAL275 | IST-Net | **84.1** | 80.9 | **61.3** | 67.6 | 79.1 | **87.4** |
> | REAL275 | SecondPose | **84.1** | **81.1** | 61.0 | **68.2** | **79.4** | 87.1 |
> | REAL275 | AG-Pose | **84.1** | **81.1** | 61.1 | 67.9 | 79.3 | 87.3 |
>
> ---
>
> **W3: The math behind the framework seems to assume that category-specific deviations are uncorrelated with the shared gradient.**
>
> - Fig. 3 reports direct empirical gradient statistics. The diagnostics in Fig. 3 are not inferred from the approximation in Sec. 4.2, but are computed directly from empirical category-wise gradients using the cosine-based interaction metrics in Eqs. (2)-(6), including category-to-category alignment, category-to-all alignment, and the derived statistics $\mu_{cc}$, $\sigma^2_{cc}$, and $N(c)$. The decomposition $g_c = u + v_c$ and the uncorrelated-deviation assumption are introduced only to provide theoretical intuition for the observed trend, rather than to compute the diagnostics themselves.
>
> - We agree that gradients can be noisy at the early stage of training and may introduce local fluctuations. However, the conclusion in Fig. 3 is based on the consistent trend across training, rather than on any single early checkpoint. Moreover, as detailed in Appendix B, especially Secs. B.1-B.2, all diagnostics are computed offline from saved checkpoints on a fixed diagnostic subset, without parameter updates, and under identical diagnostic settings across epochs. This makes the statistics directly comparable over time and reduces the influence of transient optimization noise.
>
> - We will clarify this point in the paper: Fig. 3 reports direct empirical gradient statistics, whereas the uncorrelated-deviation assumption is introduced only to provide theoretical intuition for the observed trend.

---

> > ### Author Rebuttal · Reviewer_xFUw · 2026-04-01
> >
> > Thank you for the thoughtful rebuttal, which fully addresses my questions. The justification for the math behind the framework sounds correct and provides valuable insights to leverage gradient information.  I maintain my recommendation of acceptance for this paper.

---

> > > ### Author Response · Authors · 2026-04-05
> > >
> > > Dear Reviewer,
> > >
> > > Thank you again for your thoughtful feedback and for increasing your confidence after reading our rebuttal. We are very encouraged that the discussion was helpful. In response to your comments, we made several improvements in the revision:
> > >
> > > **HouseCat6D evaluation.**
> > > - We added the missing GCE-Pose comparison. After including this method, our approach remains competitive and achieves the best performance on the main strict pose metrics under the stronger grouping setting.
> > >
> > > **Dependence on a single reference estimator.**
> > > - We additionally replaced AG-Pose with IST-Net and SecondPose as reference estimators. The results remain broadly consistent, suggesting that the routing mechanism does not depend on a specific reference model.
> > >
> > > **Gradient diagnostics (Fig. 3).**
> > > - We clarified that the statistics are computed directly from empirical category-wise gradients and evaluated offline from saved checkpoints on a fixed diagnostic subset, which improves comparability and reduces early-stage noise.
> > >
> > > Thank you again for your helpful comments and support.
> > >
> > > Best regards,
> > > Authors of Paper 11411

---

### Official Review · Reviewer_Q8Ha · 2026-03-06

**Soundness:** 1
**Presentation:** 2
**Significance:** 2
**Originality:** 2
**Overall Recommendation:** 3
**Confidence:** 5

**Summary:**

This paper tackle category-level object pose estimation. It addresses the cross-category learning problem. It first separates the categories into multiple groups based on the difficulty. Then it assigns each group with a different branch. Experiments are done on NOCS and HouseCat6D.

**Compliance With Llm Reviewing Policy:**

Affirmed.

**Final Justification:**

My two concerns are still unsolved.

1. Single-category and joint-category training under the same parameter budget does not address my concern in separating contributions of structural similarity and optimization contention. One possible ablation should be grouping categories with similar difficulty but with little strucutal smilarity.

2. The authors claim that: In practice, this proxy (difficulty) reflects multiple aspects, including category separability, training stability, and the degree of cross-category interference during joint training. This is not supported by experiments or by theory.

Although my opinion is still negative, with improved results on HouseCat6D, it would not be too bad to accept this paper. The idea of grouping is valuable to the community, but the difficulty-aware grouping strategy is not carefully designed. I raise my score to 3.

**Key Questions For Authors:**

See weakneses

**Limitations:**

NO limitations are discussed in the paper.

**Strengths And Weaknesses:**

Strengths:
1. Category-level object pose estimation is an important task.
2. The method achieves incremental performance boost on NOCS and HouseCat6D.

Weaknesses:
1. Categories with similar difficulty might not have large gradient similarity. Two categories with diffculty near to 1 (almost unable to solve) but shares no semantic similarity should not be assigned to the same branch. Simply grouping the categories using the difficulty is not appropiate. This strategy works on NOCS since NOCS only has 6 categories and the group number is only set to 3. Bottle, bowl and can happens to be similar in the geometric space. This assumption might not hold with larger datasets.
2. All ablation study and the gradient analysis are done on NOCS. HouseCat6D has 10 categories, which is more suitable to perform the experiments. Fig 3 and the ablation study should be done on HouseCat6D.
3. Assigning different group to different branches introduces more parameters, but the performance boost is too incremental. In theory, the method should bring more benefits when the category number increases. However, in HouseCat6D (the largest dataset in the paper), the performance boost on 5 degree 2 cm / 5 cm is less than 0.2%. The usage of more paramenters is not justified properly.
4. Assigning small capacity branch to high difficulty group is counter-intuitve. If the category is more difficult, it should require larger capacity to learn the correspondence pattern. Are all the design choices tuned by hand just to achieve better performance on NOCS dataset?

---

> ### Author Rebuttal · Authors · 2026-03-30
>
> **Response to Reviewer Q8Ha**
>
> We appreciate the reviewer’s helpful comments and address the concerns below.
>
> **W1: Categories with similar difficulty might not have large gradient similarity.**
>
> - We do not assume that difficulty and gradient similarity are equivalent. Gradient analysis is only used to diagnose optimization contention, while difficulty serves as a practical proxy for joint-training stability. The goal is not to group semantically similar categories, but to reduce optimization imbalance during multi-category training.
>
> - The gains are not mainly driven by geometrically similar or saturated categories. As shown below, larger improvements appear on harder categories such as laptop and mug, suggesting that the grouping mainly alleviates optimization contention.
>
> | REAL275 (5°2cm) |  |  |  |  |  |  |
> |---|---:|---:|---:|---:|---:|---:|
> | Method  | bottle | bowl | can | laptop | mug | camera |
> | AG-Pose | 68.23 | 92.21 | 81.24 | 64.61 | 37.14 | 1.65 |
> | Ours | 70.30 (+2.07) | 93.56 (+1.35) | 81.95 (+0.71) | 70.02 (+5.41) | 47.45 (+10.31) | 3.54 (+1.89) |
>
> - This behavior generalizes beyond NOCS. On HouseCat6D, our method already improves AG-Pose under the same grouping setting used on NOCS ($G=3$), and increasing the number of groups to $G=4$ further improves the stricter pose metrics.
>
>
> | HouseCat6D |  |  |  |  |  |  |
> |---|---:|---:|---:|---:|---:|---:|
> | Method | IoU50 | IoU75 | 5°2cm | 5°5cm | 10°2cm | 10°5cm |
> | AG-Pose | 76.9 | 53.0 | 21.3 | 22.1 | 51.3 | 54.3 |
> | Ours (G=3) | 77.4 | 53.9 | 22.5 | 24.3 | 52.8 | 56.8 |
> | Ours (G=4) | **79.4** | **56.3** | **25.3** | **25.9** | **54.3** | **58.9** |
>
> **W2: All ablation study and the gradient analysis are done on NOCS.**
>
> - Thank you for the helpful suggestion. We initially conducted detailed analysis on NOCS following prior works such as AG-Pose, GCE-Pose, and CleanPose. Following your suggestion, we additionally performed ablations on HouseCat6D, and the trends remain consistent with those on NOCS. Ablations on different reference estimators, branch-capacity assignments, and boundary refinement are also reported in our responses to **Reviewer HV19 (W1-W3)**.
>
> - We further analyze the effect of the number of groups $G$ on HouseCat6D. The results below show that $G=4$ gives the best overall performance, supporting that larger and more heterogeneous datasets benefit from finer grouping.
>
> | HouseCat6D |  |  |  |  |  |  |
> |---|---:|---:|---:|---:|---:|---:|
> | Groups | IoU50 | IoU75 | 5°2cm | 5°5cm | 10°2cm | 10°5cm |
> | 2 | 77.6 | 54.2 | 21.9 | 24.1 | 53.4 | 55.4 |
> | 3 | 77.4 | 53.9 | 22.5 | 24.3 | 52.8 | 56.8 |
> | 4 | **79.4** | **56.3** | **25.3** | **25.9** | 54.3 | **58.9** |
> | 5 | 78.1 | 55.6 | 23.8 | 24.5 | **55.1** | 58.1 |
>
> - Additional HouseCat6D diagnostics are provided in the anonymous supplementary material:
>   **https://anonymous.4open.science/r/aaa-0457/CC_house.pdf**
>   **https://anonymous.4open.science/r/aaa-0457/CA_VIS.pdf**
>
> **W3: Assigning different groups to different branches introduces more parameters, but the performance gain is incremental.**
>
> - The gain is not mainly from increased parameters but from alleviating cross-category optimization conflicts. To support this, we compare single-category and joint-category training under the same parameter budget on a representative difficult category, mug. Joint-category training performs better, suggesting that the gain mainly comes from improved optimization during multi-category learning.
>
> | REAL275 (Mug) |  |  |  |  |
> |---|---:|---:|---:|---:|
> | Method | 5°2cm | 5°5cm | 10°2cm | 10°5cm |
> | Single-cat. | 32.34 | 32.87 | 87.52 | 87.93 |
> | Joint-cat. | 37.14 | 37.64 | 91.76 | 92.63 |
>
> - The relatively smaller gain on HouseCat6D is mainly due to conservative grouping. For fair comparison, we initially used the same configuration as NOCS. As shown in our response to **Reviewer HV19 (W2)**, increasing the number of groups from $G=3$ to $G=4$ yields noticeably larger improvements.
>
> **W4: Assigning a small-capacity branch to the high-difficulty group is counterintuitive.**
>
> - Our goal is stable joint optimization rather than maximizing per-category capacity. In this setting, the main bottleneck is negative transfer during joint training. High-difficulty categories tend to produce noisier gradients. Assigning them overly large branches may amplify category-specific irregularities and cross-category interference, whereas lightweight branches act as an implicit regularizer.
>
> - Additional experiments **(Reviewer HV19 W2)** show that assigning larger-capacity branches to harder groups does not consistently improve performance, while the proposed asymmetric design yields more stable gains.
>
> **W5: Lack of limitations.**
>
> - A limitation of DecomPose is that its offline difficulty-based routing may become less effective when the category set grows substantially, since difficulty is only a coarse proxy for training hardness. More adaptive routing is an important future direction.

---

> > ### Author Rebuttal · Reviewer_Q8Ha · 2026-04-01
> >
> > The rebuttal addresses a large portion of my concerns. Still, my major concern in the grouping strategy is unsolved. As in HouseCat6D, the groups are (box, remote), (bottle, can, tube), (cup, teapot, glass), (cutlery, shoe). The first 3 groups still share large similarity within each group. I understand that difficulty-aware grouping does not mean to group semantically similar categories. But in both experiments in NOCS and HouseCat6D, whether the performance gain comes from structural similarity or optimization contention is not fully justified. More importantly, how the difficulty correlates with the optimization contention is not proved in theory (or in experiments, since the scale is too small).

---

> > > ### Author Response · Authors · 2026-04-01
> > >
> > > We sincerely thank the reviewer for the insightful comments. We address the main concerns below.
> > >
> > > - **Structural similarity vs. grouping criterion.**
> > >
> > > The grouping strategy is not based on structural similarity. Categories are grouped according to training difficulty estimated by the reference estimator. As a result, some groups contain structurally similar objects (e.g., bottle/can/tube), but structural similarity itself is not used as a grouping criterion. In our framework, the grouping is determined by the relative optimization difficulty observed during joint training rather than by geometric or semantic similarity.
> > >
> > > - **Additional ablation on limited structural similarity.**
> > >
> > > Following the reviewer’s suggestion, we further conduct a targeted ablation to reduce the effect of structural similarity. Specifically, on HouseCat6D, we retain six relatively heterogeneous categories, **box, remote, shoe, cutlery, teapot, and tube**, while excluding categories with more apparent geometric overlap. We then compare **no grouping**, **random grouping**, and **our difficulty-aware grouping** under the same training protocol. As shown below, difficulty-aware grouping still achieves the best performance across all metrics on this reduced subset. This result provides further evidence that the observed gains are not solely due to grouping structurally similar categories together, and supports the role of difficulty-aware grouping beyond structural similarity alone.
> > >
> > > | Method | IoU50 | IoU75 | 5°2cm | 5°5cm | 10°2cm | 10°5cm |
> > > |---|---:|---:|---:|---:|---:|---:|
> > > | No grouping | 64.0 | 49.7 | 10.3 | 12.1 | 32.7 | 36.3 |
> > > | Random grouping | 64.4 | 50.8 | 11.2 | 13.4 | 33.5 | 36.9 |
> > > | Difficulty-aware grouping | **64.6** | **51.3** | **11.9** | **14.6** | **34.8** | **37.6** |
> > >
> > > - **Distinguishing structural similarity from optimization effects.**
> > >
> > > To distinguish these two factors, we additionally compared single-category and joint-category training under the same parameter budget. If the observed improvement mainly came from structural similarity, single-category training should perform comparably because no cross-category interaction is involved. However, joint-category training consistently performs better (e.g., on the mug category), indicating that the gain mainly comes from alleviating cross-category optimization conflicts rather than exploiting structural similarity.
> > >
> > > - **Difficulty and optimization contention.**
> > >
> > > Difficulty should not be interpreted as a proxy for any single factor such as structural similarity or gradient similarity. Instead, it serves as a coarse proxy for the overall difficulty of joint optimization. In practice, this proxy reflects multiple aspects, including category separability, training stability, and the degree of cross-category interference during joint training. Categories with higher training difficulty tend to produce noisier gradients and less consistent optimization directions, which increases interference in shared representations.
> > >
> > > - **Dataset scale and empirical evidence.**
> > >
> > > We agree that current category-level pose benchmarks contain a limited number of categories. Our goal here is not to establish a strict theoretical proof of the correlation between difficulty and optimization contention, but to provide empirical evidence on standard benchmarks in this field, including NOCS and HouseCat6D. Across both datasets, the observed gradient diagnostics and ablation results consistently support the effectiveness of the proposed difficulty-aware grouping strategy.
> > >
> > > We hope this clarification addresses the reviewer’s concerns.

---

### Official Review · Reviewer_HhcP · 2026-03-12

**Soundness:** 2
**Presentation:** 3
**Significance:** 2
**Originality:** 2
**Overall Recommendation:** 4
**Confidence:** 3

**Summary:**

This paper proposes DecomPose, a difficulty-aware decomposition framework that effectively mitigates cross-category optimization contention and achieves state-of-the-art performance.

**Compliance With Llm Reviewing Policy:**

Affirmed.

**Final Justification:**

My concerns have been addressed during the rebuttal, and I recommend this paper for acceptance.

**Key Questions For Authors:**

Is the performance of DecomPose on each dataset under different metrics evaluated using the same checkpoint?

**Limitations:**

yes

**Strengths And Weaknesses:**

Strengths:
1. This paper is technically sound and well structured.
2. The gradient-based diagnostics can quantify both category-to-category conflicts and category-to-others negative transfer.

Weaknesses:
1. The performance improvements over the baselines in Table 1 appear to be marginal. Is there any statistical evidence (e.g., p-values) demonstrating a significant difference between the performance of DecomPose and the baselines?
2. It would be helpful to report the Standard Deviation or Standard Error of the Mean (SEM) for all tables and figures to better reflect the variability of the results.

---

> ### Author Rebuttal · Authors · 2026-03-30
>
> **Response to Reviewer HhcP**
>
> We appreciate the reviewer’s helpful comments and address the concerns below.
>
> ---
>
> **W1: Statistical significance of the reported improvements.**
>
> - To assess result stability, we report results from three runs with different random seeds. The first row in each block corresponds to the original submission result, where we followed the AG-Pose setting and used seed 1. During the rebuttal period, we additionally trained two more runs with different seeds.
>
> - As shown below, the run-to-run variation is consistently small for both DecomPose and AG-Pose across datasets and metrics. In particular, on the stricter pose metrics where DecomPose shows the clearest improvements, the variation of both methods is much smaller than the performance gap between them. While three runs are insufficient for a strong formal significance test, these repeated-run results provide clear evidence that the gains of DecomPose over AG-Pose are stable rather than due to incidental seed variation. We will incorporate these repeated-run statistics into the revised version.
>
> | Dataset | Seed | Method | IoU50 | IoU75 | $5^\circ2$cm | $5^\circ5$cm | $10^\circ2$cm | $10^\circ5$cm |
> | :---: | :---: | :---: | :---: | :---: | :---: | :---: | :---: | :---: |
> | REAL275 | 1 | DecomPose | 84.1 | 81.1 | 61.1 | 67.9 | 79.3 | 87.3 |
> | REAL275 | 1 | AG-Pose | 84.1 | 80.1 | 57.0 | 64.6 | 75.1 | 84.7 |
> | REAL275 | 50 | DecomPose | 84.1 | 81.2 | 61.5 | 67.7 | 79.2 | 87.6 |
> | REAL275 | 50 | AG-Pose | 84.0 | 79.8 | 57.3 | 64.2 | 74.9 | 84.5 |
> | REAL275 | 100 | DecomPose | 84.1 | 81.1 | 61.0 | 68.2 | 79.0 | 87.5 |
> | REAL275 | 100 | AG-Pose | 84.0 | 80.2 | 56.8 | 64.7 | 75.4 | 84.2 |
> | REAL275 | mean($\pm$SEM) | DecomPose | 84.1$\pm$0.0 | 81.1$\pm$0.0 | 61.2$\pm$0.2 | 67.9$\pm$0.1 | 79.2$\pm$0.1 | 87.5$\pm$0.1 |
> | REAL275 | mean($\pm$SEM) | AG-Pose | 84.0$\pm$0.0 | 80.0$\pm$0.1 | 57.0$\pm$0.1 | 64.5$\pm$0.2 | 75.1$\pm$0.1 | 84.5$\pm$0.1 |
> | CAMERA25 | 1 | DecomPose | 94.3 | 92.7 | 80.7 | 84.5 | 87.8 | 92.8 |
> | CAMERA25 | 1 | AG-Pose | 94.2 | 92.5 | 79.5 | 83.7 | 87.1 | 92.6 |
> | CAMERA25 | 50 | DecomPose | 94.3 | 92.7 | 80.8 | 84.3 | 87.9 | 92.8 |
> | CAMERA25 | 50 | AG-Pose | 94.2 | 92.4 | 79.3 | 83.8 | 87.2 | 92.3 |
> | CAMERA25 | 100 | DecomPose | 94.3 | 92.6 | 80.8 | 84.6 | 87.6 | 92.7 |
> | CAMERA25 | 100 | AG-Pose | 94.3 | 92.5 | 79.5 | 83.5 | 87.4 | 92.4 |
> | CAMERA25 | mean($\pm$SEM) | DecomPose | 94.3$\pm$0.0 | 92.7$\pm$0.0 | 80.8$\pm$0.0 | 84.5$\pm$0.1 | 87.8$\pm$0.1 | 92.8$\pm$0.0 |
> | CAMERA25 | mean($\pm$SEM) | AG-Pose | 94.2$\pm$0.0 | 92.5$\pm$0.0 | 79.4$\pm$0.1 | 83.7$\pm$0.1 | 87.2$\pm$0.1 | 92.4$\pm$0.1 |
> | HouseCat6D | 1 | DecomPose | 79.4 | 56.3 | 25.3 | 25.9 | 54.3 | 58.9 |
> | HouseCat6D | 1 | AG-Pose | 76.9 | 53.0 | 21.3 | 22.1 | 51.3 | 54.3 |
> | HouseCat6D | 50 | DecomPose | 79.5 | 56.2 | 25.1 | 25.6 | 54.0 | 58.7 |
> | HouseCat6D | 50 | AG-Pose | 76.5 | 53.3 | 21.5 | 22.3 | 51.2 | 54.2 |
> | HouseCat6D | 100 | DecomPose | 79.3 | 56.5 | 25.7 | 26.1 | 54.4 | 59.2 |
> | HouseCat6D | 100 | AG-Pose | 76.8 | 53.2 | 20.9 | 22.4 | 51.2 | 54.6 |
> | HouseCat6D | mean($\pm$SEM) | DecomPose | 79.4$\pm$0.1 | 56.3$\pm$0.1 | 25.4$\pm$0.2 | 25.9$\pm$0.1 | 54.2$\pm$0.1 | 58.9$\pm$0.1 |
> | HouseCat6D | mean($\pm$SEM) | AG-Pose | 76.7$\pm$0.1 | 53.2$\pm$0.1 | 21.2$\pm$0.2 | 22.3$\pm$0.1 | 51.2$\pm$0.0 | 54.4$\pm$0.1 |
>
> ---
>
> **W2: Variability reporting with SEM in tables and figures.**
>
> - Thank you for this helpful suggestion. As shown in W1, we report SEM for the main quantitative results on all three datasets to reflect result stability across repeated runs.
>
> - For the analysis figures, we report SEM for the category-to-category gradient similarity results in (**https://anonymous.4open.science/r/aaa-0457/CC_SEM.pdf**). The trends remain consistent with Fig. 3 in the submission version. For the localization of optimization contention and the category-to-all gradient similarity curves, our goal is mainly to show overall training trends, so we do not add SEM to these plots. We will clarify this more explicitly in the revised text.
>
> ---
>
> **W3: Whether the same checkpoint is used across datasets and metrics.**
>
> - Following prior works such as AG-Pose and CleanPose, CAMERA25 and REAL275 are evaluated using the same checkpoint trained under the standard joint CAMERA+REAL protocol. HouseCat6D is trained and evaluated separately on its own split, and therefore uses a different checkpoint. We will clarify this protocol in the revised version.

---

> > ### Author Rebuttal · Reviewer_HhcP · 2026-04-01
> >
> > Thanks for the rebuttal. I will raise the score to 4.

---

> > > ### Author Response · Authors · 2026-04-05
> > >
> > > Dear Reviewer,
> > >
> > > Thank you very much for your thoughtful follow-up and for taking the time to carefully read our rebuttal. We truly appreciate your positive feedback and are encouraged to know that our responses have adequately addressed your concerns.
> > >
> > > We are especially grateful for your recognition that the issues regarding statistical stability, SEM reporting, and evaluation protocol clarification have been resolved. Your suggestions were very helpful in improving both the clarity and completeness of the paper, and we will incorporate these clarifications into the final revision.
> > >
> > > Thank you again for your time, support, and constructive feedback.
> > >
> > > Best regards,
> > > Authors of Paper 11411

---

### Official Review · Reviewer_HV19 · 2026-03-13

**Soundness:** 3
**Presentation:** 4
**Significance:** 3
**Originality:** 3
**Overall Recommendation:** 4
**Confidence:** 5

**Summary:**

This paper presents DecomPose, a framework designed to mitigate gradient conflicts arising from geometric heterogeneity in category-level 6D pose estimation. By employing gradient-based diagnostics, the authors identify the correspondence module as the primary source of contention and propose a difficulty-aware routing mechanism with asymmetric branching. This approach effectively decouples incompatible optimization signals, achieving state-of-the-art results across multiple benchmarks. Overall, the authors focus on a broad topic regarding multi-category optimization. This manuscript's primary contribution comprises a novel gradient-based diagnostic and a difficulty-aware decoupling architecture.

**Compliance With Llm Reviewing Policy:**

Affirmed.

**Key Questions For Authors:**

See weaknesses

**Limitations:**

yes

**Strengths And Weaknesses:**

**Strengths**

- Novel Perspective: Instead of just improving feature extraction, the paper focuses on the dynamics of joint optimization, bringing insights from Multi-Task Learning to pose estimation.
- Strong Theoretical Foundation: The use of the "module-wise heterogeneity ratio" provides a solid theoretical basis for identifying where decoupling is needed.
- Efficient Design: The minimal decoupling of only the correspondence module keeps the framework efficient while yielding significant gains.
- Extensive Validation: Consistent improvements across three benchmarks, including the challenging HouseCat6D dataset.

**Weaknesses & Questions**

- The grouping depends on scores from a pre-computed reference estimator. How does the system perform if no reliable reference estimator is available? Is the static routing robust enough for datasets with a massive number of categories?

- Complex categories are assigned lightweight branches to serve as implicit regularization, which is a bit counterintuitive. Since complex shapes usually require more parameters to model, could the authors provide more empirical evidence showing that giving complex categories high-capacity branches leads to worse negative transfer or optimization instability?

- Possible boundary refinement overhead. Algorithm 1 involves a "Boundary Refinement" for boundary categories. Does this process scale well if the category set is significantly larger?

---

> ### Author Rebuttal · Authors · 2026-03-30
>
> **Response to Reviewer HV19**
>
> We appreciate the reviewer’s helpful comments and address the concerns below.
>
> ---
>
> **W1: Robustness to the reference estimator and scalability to larger category sets.**
>
> - We replaced AG-Pose with weaker proxy estimators, IST-Net and SecondPose, when constructing the routing groups. As shown below, performance remains largely stable on both HouseCat6D and REAL275. Although AG-Pose gives the best overall results, the differences among the three reference models are small, suggesting that the routing does not rely on a uniquely strong reference estimator.
>
> | Dataset | Reference | IoU50 | IoU75 | 5°2cm | 5°5cm | 10°2cm | 10°5cm |
> |---|---|---:|---:|---:|---:|---:|---:|
> | HouseCat6D | IST-Net | 78.2 | 54.9 | 23.9 | 24.7 | 53.6 | 57.8 |
> | HouseCat6D | SecondPose | 79.1 | **56.5** | 24.7 | 25.4 | 53.8 | 58.6 |
> | HouseCat6D | AG-Pose | **79.4** | 56.3 | **25.3** | **25.9** | **54.3** | **58.9** |
> | REAL275 | IST-Net | **84.1** | 80.9 | **61.3** | 67.6 | 79.1 | **87.4** |
> | REAL275 | SecondPose | **84.1** | **81.1** | 61.0 | **68.2** | **79.4** | 87.1 |
> | REAL275 | AG-Pose | **84.1** | **81.1** | 61.1 | 67.9 | 79.3 | 87.3 |
>
> - We also compare three routing strategies: random routing, proxy routing without boundary refinement, and proxy routing with boundary refinement. On both HouseCat6D and REAL275, proxy-based routing consistently outperforms random routing, and boundary refinement further improves performance.
>
> | Dataset | Method | IoU50 | IoU75 | 5°2cm | 5°5cm | 10°2cm | 10°5cm |
> |---|---|---:|---:|---:|---:|---:|---:|
> | HouseCat6D | Random | 76.2 | 52.1 | 20.7 | 21.6 | 51.3 | 55.1 |
> | HouseCat6D | Proxy w/o BR | 78.6 | 54.3 | 22.9 | 23.7 | 52.6 | 57.2 |
> | HouseCat6D | Proxy w/ BR | **79.4** | **56.3** | **25.3** | **25.9** | **54.3** | **58.9** |
> | REAL275 | Random | **84.1** | 80.3 | 59.2 | 66.4 | 76.2 | 86.4 |
> | REAL275 | Proxy w/o BR | 84.0 | 80.6 | 60.1 | 66.9 | 77.4 | 86.5 |
> | REAL275 | Proxy w/ BR | **84.1** | **81.1** | **61.1** | **67.9** | **79.3** | **87.3** |
>
> ---
>
> **W2: Evidence that assigning complex categories high-capacity branches can worsen optimization.**
>
> - Our goal is not to maximize per-category fitting capacity in isolation, but to improve optimization stability under joint training. Empirically, assigning the hardest group to a higher-capacity branch is not more effective in our setting. As shown below, H/H/L/L consistently outperforms H/H/H/H on HouseCat6D.
>
> - We further conducted a diagnostic analysis similar to Fig. 3(c), comparing gradient-direction dynamics under H/H/H/H and H/H/L/L. The corresponding plots are available in the anonymous supplementary material:  **https://anonymous.4open.science/r/aaa-0457/CAP_VIS.pdf**
>
> - The results show that H/H/L/L yields more stable gradient evolution, supporting our interpretation that lightweight branches can act as an implicit regularizer for difficult categories.
>
> | HouseCat6D |  |  |  |  |  |  |
> |---|---:|---:|---:|---:|---:|---:|
> | Branch | IoU50 | IoU75 | 5°2cm | 5°5cm | 10°2cm | 10°5cm |
> | L/L/L/L | 76.5 | 53.1 | 22.3 | 23.8 | 53.1 | 55.6 |
> | H/H/H/H | 76.7 | 55.2 | 23.1 | 24.6 | **54.5** | 56.2 |
> | L/L/H/H | 77.1 | 54.3 | 22.9 | 24.1 | 53.7 | 57.1 |
> | H/H/L/L | **79.4** | **56.3** | **25.3** | **25.9** | 54.3 | **58.9** |
>
> ---
>
> **W3: Scalability of Boundary Refinement.**
>
> - Boundary Refinement (BR) is an offline routing refinement step applied only to a small subset of ambiguous boundary categories and executed once during group construction.
>
> - To examine scalability, we evaluated BR on HouseCat6D. As shown below, BR consistently improves performance on both HouseCat6D and REAL275.
>
> - Additional diagnostic results are available in the anonymous supplementary material:
> **https://anonymous.4open.science/r/aaa-0457/CC_house.pdf**
> **https://anonymous.4open.science/r/aaa-0457/CA_VIS.pdf**
>
> | Dataset | Method | IoU50 | IoU75 | 5°2cm | 5°5cm | 10°2cm | 10°5cm |
> |---|---|---:|---:|---:|---:|---:|---:|
> | HouseCat6D | Proxy w/o BR | 78.6 | 54.3 | 22.9 | 23.7 | 52.6 | 57.2 |
> | HouseCat6D | Proxy w/ BR | **79.4** | **56.3** | **25.3** | **25.9** | **54.3** | **58.9** |
> | REAL275 | Proxy w/o BR | 84.0 | 80.6 | 60.1 | 66.9 | 77.4 | 86.5 |
> | REAL275 | Proxy w/ BR | **84.1** | **81.1** | **61.1** | **67.9** | **79.3** | **87.3** |

---

### Decision · Program_Chairs · 2026-04-30

**Decision:**

Accept (regular)

**Comment:**

This paper addresses an important problem in category-level 6D object pose estimation by reframing performance degradation in joint multi-category training as a problem of cross-category optimization contention. Reviewers found the paper technically solid, well written, and notably original in its use of gradient-based diagnostics to localize contention and motivate a targeted decomposition of the correspondence module. The experimental results across REAL275, CAMERA25, and HouseCat6D were viewed as strong overall, and the proposed design was considered both effective and relatively efficient. Three of the four reviewers recommended acceptance, and after discussion the overall reviewer consensus supports acceptance.

The main concerns were about whether difficulty is an adequately justified routing proxy, whether gains could partly reflect structural similarity rather than reduced optimization interference, the reliance on a reference estimator, missing baseline coverage on HouseCat6D, and the need for stronger evidence on result stability. The authors addressed these concerns substantively by adding the missing GCE-Pose comparison, reporting repeated-run statistics and SEM, showing robustness to alternative reference estimators, expanding HouseCat6D ablations, and clarifying that the difficulty signal is intended as a practical coarse heuristic rather than a complete theoretical characterization. One reviewer remained somewhat unconvinced about the grouping rationale, but even that reviewer acknowledged the idea’s value and softened their stance after the additional evidence.

For the camera-ready version, the authors should make sure the paper clearly states the limitations of offline difficulty-based routing, tones claims to match the empirical evidence, and incorporates the added experimental clarifications from the rebuttal. Overall, this is a useful and technically sound contribution that should be of interest to the community.